# Tracking neurons across days with high-density probes

Enny H. van Beest ⓘ [1,4] ✉, Célian Bimbard ⓘ [1,4] ✉, Julie M. J. Fabre[2], Sam W. Dodgson ⓘ [2], Flóra Takács ⓘ [3], Philip Coen[1], Anna Lebedeva ⓘ [3], Kenneth D. Harris ⓘ [2] & Matteo Carandini ⓘ [1]

Neural activity spans multiple time scales, from milliseconds to months. Its evolution can be recorded with chronic high-density arrays such as Neuropixels probes, which can measure each spike at tens of sites and record hundreds of neurons. These probes produce vast amounts of data that require different approaches for tracking neurons across recordings. Here, to meet this need, we developed UnitMatch, a pipeline that operates after spike sorting, based only on each unit's average spike waveform. We tested UnitMatch in Neuropixels recordings from the mouse brain, where it tracked neurons across weeks. Across the brain, neurons had distinctive inter-spike interval distributions. Their correlations with other neurons remained stable over weeks. In the visual cortex, the neurons' selectivity for visual stimuli remained similarly stable. In the striatum, however, neuronal responses changed across days during learning of a task. UnitMatch is thus a promising tool to reveal both invariance and plasticity in neural activity across days.

Neural activity spans a multitude of time scales, from the milliseconds that separate spikes to the hours, days or months that characterize learning, memory or aging. Changes at these longer time scales can be studied with two-photon imaging, where the same neurons can be visually tracked across days[1–5]. However, imaging methods lack the fast time scales and are hard to deploy in deep brain regions. To cover all time scales in all brain regions, the ideal method is chronic electrophysiology.

Recordings with chronic electrodes reveal units (putative neurons) with consistent spike waveforms across days[6–21]. This constancy indicates that the units track the same neurons over time, particularly when the spikes are measured at multiple locations with stereotrodes[15], tetrodes[13,14,18,22–27], microwire bundles[28,29], silicon probes[19,30], polymer arrays[20] or Neuropixels probes[31]. The latter are readily implanted chronically[31–36] and yield hundreds of potentially matchable neurons across days. In addition, their geometry and density allow for correction of electrode drift[31,37].

The current methods for matching neurons across days, however, cannot process the vast amounts of data produced by sequences of recordings with high-density probes such as Neuropixels. For example, an established method relies on concatenating two recordings and spike sorting the resulting file[30,31]. This method can work well for pairs of recordings but becomes unwieldy for longer sequences. It does not scale to the dozens of recordings that may be obtained across weeks or months.

To solve this problem, we developed a pipeline called UnitMatch, which operates after spike sorting. Before applying UnitMatch, the user spike sorts each recording independently using their preferred algorithm. UnitMatch then deploys a naive Bayes classifier on the units' average waveform in each recording and tracks units across recordings, assigning a probability to each match.

We tested UnitMatch on sequences of Neuropixels recordings from multiple regions of the mouse brain and found that it reliably tracked neurons across weeks. Its performance compares well to the concatenated method and to curation by human experts, while being much faster and applicable to longer sequences of recordings.

Because UnitMatch relies only on each unit's spike waveform, and not on any functional properties, it can be used to test whether these

[1]UCL Institute of Ophthalmology, University College London, London, UK. [2]UCL Queen Square Institute of Neurology, University College London, London, UK. [3]Sainsbury Wellcome Centre, University College London, London, UK. [4]These authors contributed equally: Enny H. van Beest, Célian Bimbard. ✉e-mail: e.beest@ucl.ac.uk; c.bimbard@ucl.ac.uk

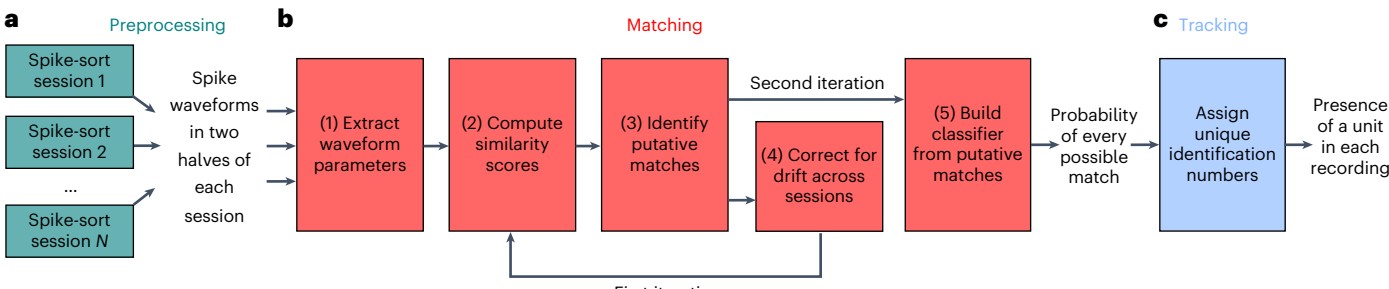

**Fig. 1 | The UnitMatch pipeline. a**, Preprocessing. The user performs spike sorting on each recording session and computes the average spike waveform of each unit in each half of each recording. **b**, Matching. UnitMatch extracts key parameters from each waveform (step 1) and uses them to compute similarity scores for each pair of units across all pairs of recordings (step 2). It then uses within-day cross-validation to identify a similarity score threshold for putative matches (step 3). It corrects for drift across recordings (step 4), and repeats steps 2 and 3 to recompute the putative matches. Finally, it builds probability distributions for the similarity scores for putative matches and feeds them to a classifier to assign a probability to every possible match across all pairs of recordings (step 5). The result is a probability for every pair of neurons across all recordings to be a match. **c**, Tracking. UnitMatch uses the probabilities output by the previous stage to track individual units across multiple sessions.

properties change over time. Indeed, while units can maintain firing properties such as inter-spike interval (ISI) distribution[10–12,19,20,28,29] and sensory, cognitive or motor correlates[11,13–15,24,28,29,31,38], the stability of these properties cannot be assumed. In fact, it is often the question being investigated[6,7,19,21–23,25,27,28,38–40].

We examined properties of neurons such as ISI distributions, correlations with other neurons and responses to visual stimuli (for neurons in visual cortex). These distinctive properties remained remarkably stable. We also used UnitMatch to characterize the changes of neural representations in the striatum during learning. These results indicate that UnitMatch can track neural activity in multiple brain regions across long sequences of recordings.

## Results

UnitMatch takes as input the spike waveforms of units that have been spike-sorted independently across recording sessions, averaged across each half of each session (Fig. 1a), and operates in two stages. The first stage, 'matching', produces the probability that each unit in a recording matches a unit in another recording (Fig. 1b). The second stage, 'tracking', produces a matrix of indices that track a unit across recordings (Fig. 1c). Below we describe these stages and illustrate them on a large body of data obtained in our laboratory. Our description here is qualitative; the relevant equations are referenced and provided in Methods.

### Preprocessing

Before running UnitMatch, the users record neural activity in multiple sessions, and use their preferred software to spike-sort each recording independently. For each recording, the output of the spike-sorting software is then used to extract for each unit a file with the average spatiotemporal spike waveform in the first and in the second half of each recording. These files have no information on individual spikes.

To develop and test UnitMatch, we used 1,350 recordings performed over multiple days (up to 235 days from a single probe) in mice implanted with chronic Neuropixels probes[31,35,41] in multiple brain regions including cortex, hippocampus, striatum and superior colliculus (Extended Data Table 1). Each recording session was individually spike-sorted with Kilosort[42], which provides drift correction within each session[31]. After spike sorting, we used a set of quality measures[43] to select $25.2 \pm 10.2\%$ (mean ± standard deviation, $n = 1,350$ recording sessions across 25 mice) units that were well isolated and distinct from noise (Extended Data Fig. 1).

### Extraction of waveform parameters

High-density recording arrays such as Neuropixels probes sample the spikes of a unit at many recording sites (Fig. 2a), revealing the unit's

characteristic spatiotemporal waveform (Fig. 2b). The amplitude of the waveform peaks at a maximum site and decays with distance from that site (Fig. 2b,c). UnitMatch fits this decay with an exponential function and obtains the distance $d_{10}$ at which the amplitude reaches 10% of the maximum (Fig. 2c). In the example recordings, this value ranged between 30 and 95 µm (95% confidence interval; Fig. 2d). For each unit, UnitMatch considers the recording sites closer than $d_{10}$ (but at most 150 µm away) from the maximum site. In our data, this typically resulted in 6–24 sites arranged in two columns (for example, Fig. 2b).

For each unit and each of its two averaged waveforms, UnitMatch uses the spatiotemporal spike waveform measured at the selected recording sites to extract seven attributes:

- The spatial decay (Fig. 2c and equation (8)).
- The weighted-average waveform (Fig. 2e and equation (9)) obtained by averaging across sites, weighted by the proximity of each site to the maximum site.
- The amplitude of the weighted-average waveform (Fig. 2e and equation (10)).
- The average centroid (Fig. 2f and equation (6)), defined as the average position weighted by the maximum amplitude on each recording site.
- The trajectory of the spatial centroid from 0.2 ms before the peak to 0.5 ms after the peak (Fig. 2f and equation (4)).
- The distance traveled at each time point (Fig. 2f).
- The travel direction of the spatial centroid at each time point (Fig. 2f and equation (5)).

### Computation of similarity scores

After extracting these spatiotemporal waveform parameters, Unit-Match compares them for every pair of waveforms within and across all recordings, to obtain six similarity scores:

- Decay similarity ($D$; equation (14));
- Waveform similarity ($W$; equation (18));
- Amplitude similarity ($A$; equation (13));
- Centroid similarity ($C$; equation (20));
- Volatility similarity ($V$; stability of the difference between centroids, equation (23));
- Route similarity ($R$; similarity of the trajectory, equation (24)).

Each similarity score is scaled between 0 and 1, with 1 indicating the highest similarity. Finally, we also average the individual scores to compute a total similarity score $T$.

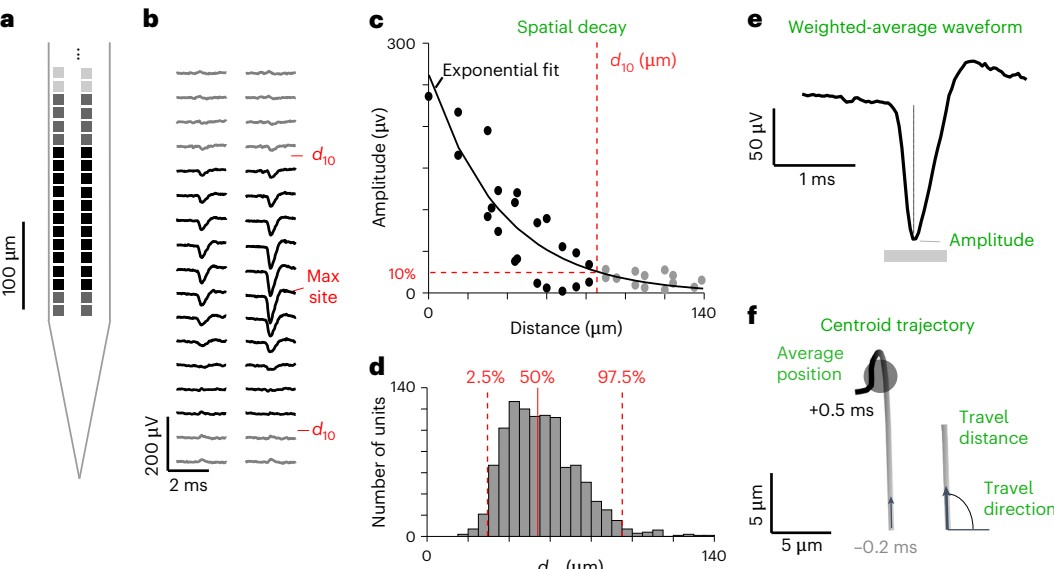

**Fig. 2 | Extracting spike waveform parameters. a**, The bottom of one shank and its recording sites. **b**, The average spike waveform for an example unit, in the 22 recording sites marked in black in **a** and in 12 adjacent sites (gray). **c**, The amplitude of the waveform as a function of distance to 'max site' for the example unit. Using an exponential decay fit (curve), we defined the distance $d_{10}$ at which the amplitude drops to 10% of the maximum. Spatial decay is computed from the slope of the amplitude decrease over distance. **d**, The distribution of $d_{10}$ for all units in two example recordings, showing the median (solid line) and the 95% confidence interval (dashed lines). **e**, The weighted-average waveform for the example unit in **b** and **c**, computed by giving larger weight to sites near the maximum site. The unit's amplitude is taken from this weighted-average waveform. **f**, The centroid trajectory of the example waveform from 0.2 ms before the peak (bottom) to 0.5 ms after the peak (top), showing the average centroid (circle). The travel direction and distance are calculated at each time point.

To gain an intuition for these scores, consider their values for two example pairs of units. The first example involves two neighboring but distinct units recorded on the same day (Fig. 3a,b). Because they are neighbors, they have high centroid similarity $C$. However, their spike waveforms are different (low value of $W$) and so are their spatial decays (low value of $D$) and routes (low value of $R$). As a result, the total similarity score $T$ is well below 1 (Fig. 3c). Conversely, in the case of a single unit recorded in two different days, we observed similar waveforms and trajectories (Fig. 3d,e), with high values of most similarity scores and, consequently, a total similarity score $T$ near the maximal value of 1 (Fig. 3f).

### Identification of putative matches

As expected, the total similarity score $T$ is generally high when applied to the same unit recorded across the two halves of a single recording (Fig. 3g, main diagonal). Indeed, the values of $T$ measured for the same unit across two halves were markedly higher than those measured across neighboring units (Fig. 3h, green versus blue curves).

UnitMatch leverages this difference to define a threshold as the value of $T$ where the proportion of pairs from the same unit exceeds the proportion of pairs from neighboring units (Fig. 3h and equation (28)). It then applies the threshold to the distribution of $T$ across days (Fig. 3i). The pairs of units recorded across days with values of $T$ beyond this threshold are putative matches.

### Correction for drift across sessions

Modern spike-sorting algorithms (including the one we used[42]) correct for electrode drift within a session but naturally cannot correct for drift across sessions that are separately sorted. This can lead to larger values of total similarity score $T$ measured within a day than across days.

To adjust for this difference, UnitMatch fits a Gaussian function to the two distributions of total similarity scores (within and across days) and uses the fit to equalize their means and again identify putative matches. For these putative matches, it computes the median centroid

displacement and uses it to rigidly transform all parameters affected by position. It then repeats the previous two steps, thus finding a more robust set of putative matches. The results we have shown (Fig. 3) are from after drift correction.

### Building a classifier from putative matches

Having used the total similarity score $T$ to identify putative matches across all pairs of recordings (Fig. 3i), UnitMatch goes back to the individual similarity scores and uses their distributions to train a classifier. There are two types of pair: putative matches ($T >$ threshold) and putative nonmatches ($T <$ threshold; Fig. 3i). The distributions of the six similarity scores for these pairs differ substantially (Fig. 3j). Based on these distributions, we defined a naive Bayes classifier, which takes as input the values of the six similarity scores for two spike waveforms and outputs the 'match probability': the posterior probability of the two waveforms coming from the same unit (equation (29)).

This classifier correctly identified the same unit within a day with match probabilities close to 1 (Fig. 3k, main diagonal and Fig. 3l, green curve). On the contrary, the matching probabilities of neighboring units were close to 0 (Fig. 3l, blue). Across days, most pairs of waveforms are expected to come from different neurons, which is reflected in a large portion of match probabilities close to 0 (Fig. 3m). However, a fraction of pairs had a match probability close to 1. These matches reflect units tracked across days.

### Performance metrics

We first evaluated the performance of UnitMatch on waveforms obtained within days and confirmed that it is overall accurate while being conservative. We applied UnitMatch to units recorded in two halves of a single recording session, which were assessed to be the same across the two halves by the spike-sorting algorithm (here, Kilosort[42]). Consistent with the way the classifier was trained, UnitMatch tended to agree with the algorithm on these within-day matches (Extended Data Fig. 2a). Disagreements were rare: in ten recordings spike-sorted

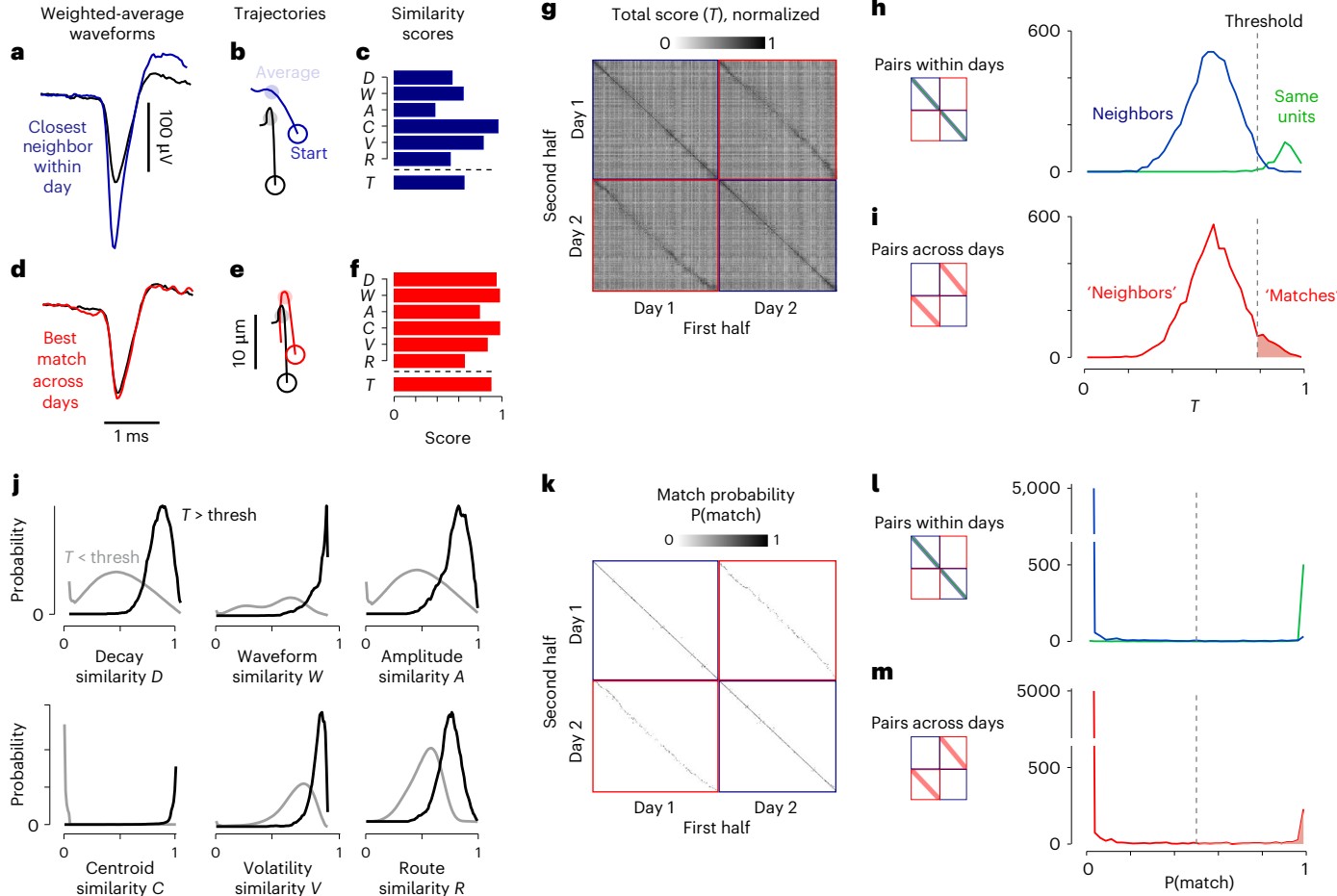

**Fig. 3 | Computing similarity scores and setting up the classifier. a**, The average weighted waveform for the example unit in Fig. 2 (black) and for a different unit (the nearest physical neighbor) from the same recording (blue). **b**, Centroid trajectories for the two units. The average position is shown by the shaded circle on the trajectory. **c**, The six similarity scores between the two units and their average, the total similarity score *T*. **d**–**f**, The same as **a** (**d**), **b** (**e**) and **c** (**f**), comparing the example unit (black) with the most similar unit across days (red), which was very likely to be the same neuron. **g**, The total similarity score for all pairs of units within days (blue squares) and across days (red squares), for an example pair of recordings, showing the first half of each recording (columns) versus the second half (rows). The data are sorted by shank, and then by depth on the shank. **h**, The distribution of total similarity scores in the two halves of a recording day, shown for the same units across the two halves (green) and for other neighboring units (centroid <50 μm away; blue). **i**, The same as **h**, but for units measured across days (red) after drift correction. The threshold (thresh) for putative matching (dashed line) depends on the number of units and of recordings. **j**, The probability densities of each similarity score, for putative matches (black) and for putative nonmatches (gray). **k**, Match probability P(match) computed by the naive Bayes classifier trained with the probability distributions in **i**. Format as in **g**. **l**, The distribution of match probabilities across two halves of the same day for same units (green) and neighbors (blue). Format as in **h**. **m**, The same as **l**, but for units recorded across days (red). If probability is >0.5, UnitMatch defines a pair as a match.

individually, UnitMatch found 0.2 ± 0.1% (median ± median absolute deviation (m.a.d.)) unexpected matches and 4.2 ± 0.8% unexpected nonmatches (Extended Data Fig. 2a). These few disagreements might represent false positives and false negatives by UnitMatch or mistakes by the spike-sorting algorithm.

From the maximum possible number of units recorded across two consecutive days, UnitMatch found 42 ± 19% (median ± m.a.d., *n* = 339 pairs of days) of units to be a match. Reassuringly, when we applied UnitMatch to acute recordings, where the probe was reinserted daily and had negligible chance of finding the same unit, finding a match was rare (1.9 ± 3.7%, *n* = 21 pairs of consecutive days; Wilcoxon rank sum comparing chronic and acute: *P* < 10⁻¹¹; Extended Data Fig. 2b).

Next, we compared the performance of UnitMatch with spike sorting performed on concatenated recordings (as if they were obtained in a single session)[30,31]. Running UnitMatch on the output of Kilosort[42] on these concatenated recordings yielded similar levels of unexpected matches (0.3 ± 0.3%, *N* = 5 mice, each two recordings) and nonmatches (5.9 ± 1.9%) within days as when the recordings were sorted separately

(Extended Data Fig. 2a). Across days, 29.5 ± 16.4% of units that were identified as the same unit by Kilosort were not identified as matches by UnitMatch (Extended Data Fig. 2c).

Given the substantial difference between UnitMatch and Kilosort, we asked which method agreed more closely with human curation, where the majority of six experts had to agree on a pair being a match (Extended Data Fig. 3). We found that UnitMatch performed more similarly to manual curation than the sorting on stitched recordings did (Extended Data Fig. 2c,d). Sorting the stitched recordings with Kilosort tended to overestimate the number of matches across recordings, specifically for noisier datasets. The agreement between UnitMatch and manual curation is reassuring because the latter is often regarded more highly than automated spike sorting. However, neither can be considered ground truth.

Finally, we examined whether UnitMatch is biased toward tracking units with specific waveform and firing properties (Extended Data Fig. 4). We found that tracking could be predicted by several of these properties. As expected, the number of spikes and peak amplitude

were highly predictive. In addition, we found some predictive power in waveform duration (units with thinner spike were slightly less likely to get a match) and number of peaks (units with more peaks were slightly less likely to get a match). These features may point the way toward future improvements of the algorithm.

### Validation with stable functional properties

A more reliable estimate of UnitMatch's abilities can be found by assessing the neurons' functional 'fingerprint' (that is, pattern of activity). If this fingerprint turns out to be both distinctive across neurons and stable across days, one can conclude that the tracking algorithm performed well. We found functional fingerprints to be both distinctive and remarkably stable, thus validating UnitMatch's performance.

We considered three possible fingerprints: a unit's distribution of ISIs[8,10–12,19,20,24,28,29], its population coupling (the instantaneous correlation of its firing rate with that of the other units recorded at the same time[11,24,44,45]) and its response to a large set of visual stimuli (for units in visual cortex[28,29,31,38]). We considered each possible pair of days independently and computed the similarity of the functional fingerprints across days for matching and nonmatching pairs.

The histograms of ISIs of tracked units remained highly consistent across days. This distribution is often considered to be distinctive and stable: it has been used as a feature to track units across days[10–12,19,28,29] or as a diagnostic of this tracking[8,24]. Accordingly, the ISI histograms were typically different for neighboring units recorded within a day but similar for units matched across days (Fig. 4a). In an example mouse, the ISI distributions of pairs of units matched across days tended to be highly correlated, nearly as highly as the ISI histograms of the same units measured in the same session (Fig. 4b). Conversely, the ISI histograms of units that UnitMatch defined as different had much lower correlations (Fig. 4b).

Indeed, the correlation between the ISI histograms of a pair of units was highly predictive of whether the units were matched, even for recordings performed 6 months apart. We characterized the separation of the distributions of the correlations of matched and nonmatched pairs by computing the receiver operating characteristic (ROC) curve (Fig. 4c). The area under the curve (AUC) for the example pair of days was 0.95, almost as high as the value measured within days (0.97). Similar values were seen when increasing the number of days between recordings (Fig. 4d and Extended Data Fig. 5a,b for a breakdown of all pairs of days) and across all mice (Fig. 4e and Extended Data Fig. 5 for example mice). On average, the AUC was 0.88 ± 0.01 across days (0.94 ± 0.01 within days, mean ± standard error of the mean (s.e.m.), $n = 16$ mice) and decayed slowly with each additional day between recordings (−0.001 ± 0.008, median ± m.a.d., $n = 16$ mice). For the example mouse, the AUC remained at 0.82 after 183 days.

We then examined population coupling and found it to be also remarkably consistent across days[11,24,44,45]. This measure provided a distinctive 'fingerprint' that was highly correlated both within and across days (Fig. 4f,g), and not for neighboring units. The discriminability of this measure was particularly high, with AUC indices close to 1 (0.98

across days versus 0.98 within days; Fig. 4h), indicating that the pairs found by UnitMatch were indeed highly likely to be the same across days. Again, this held true across mice (0.92 ± 0.01 versus 0.96 ± 0.01 across mice, mean ± s.e.m., $n = 16$ mice) and even across weeks and months (slope of −0.006 ± 0.010, median ± m.a.d., $n = 16$ mice), suggesting that the correlation patterns of the population of neurons were highly stable over time (Fig. 4i,j and Extended Data Fig. 5). For the example mouse, the AUC was still 0.82 after 183 days. This fingerprint, along with the ISI histograms, can be used in any region of the brain since it does not depend on responses to stimuli.

Finally, units in visual cortex that were tracked across days also typically showed consistent responses to visual stimuli. Neurons in mouse visual cortex give distinctive responses to natural images, and these responses can remain constant across days[28,29,31,38]. Consistent with this, a typical unit matched by UnitMatch across days gave similar responses to natural images on each day (Fig. 4k,l). In the example mouse, it yielded AUCs of 0.95 across days versus 0.96 within days (Fig. 4m). Similar results were seen across mice, with AUCs of 0.85 ± 0.02 versus 0.90 ± 0.03 (mean ± s.e.m., $n = 9$ mice). This held true even with long intervals between recordings (slope of −0.006 ± 0.010, median ± m.a.d., $n = 9$ mice; Fig. 4n,o and Extended Data Fig. 5). For the example mouse, the AUC was still 0.75 after 183 days.

### Comparison with other methods

The stability of functional properties offered another opportunity to compare the performance of UnitMatch to the established method of running the spike-sorting algorithm on stitched recordings. We applied both methods on a sequence of four recording sessions and evaluated their accuracy using functional properties. In line with the earlier curation results, functional validation showed a larger overlap with the output of UnitMatch than with spike sorting the concatenated data. Indeed, the AUC for distinguishing matches versus nonmatches with functional similarity scores was generally higher for UnitMatch than for Kilosort, especially when recordings were many days apart (Extended Data Fig. 2e,f).

We also compared the performance of UnitMatch with that of a recently published tracking algorithm[46] based on Earth Mover's Distance (EMD). We tested both algorithms on the first recordings from five mice recorded in our laboratory (Extended Data Fig. 6a). Overall, UnitMatch had a larger hit rate than the EMD method for within-day performance, and a more consistently low false positive rate. We compared the ability to track neurons across 22 recordings of the mouse used as an example in the EMD paper[46] (Extended Data Fig. 6b). UnitMatch did this tracking in 25 min, whereas the EMD algorithm took 8 h. Leveraging the high stability of ISI histograms across recordings, we computed AUC values for matches versus nonmatches for both algorithms. Matches made by UnitMatch had significantly higher AUC values (paired $t$-test; $t(20) = 4.57$, $P = 0.0002$). Overall, this supports UnitMatch as being a fast and well-performing algorithm compared with state-of-the-art methods.

---

**Fig. 4 | Validation with stable functional properties. a**, Histogram of ISIs of an example unit for both halves of the recording (top), of one of its neighbors (middle) and of the example unit's match on the next day (bottom). **b**, The distribution of the correlation in ISI histograms for pairs of waveforms coming from the same unit (green) or different units (blue) within days and matches across days (red). Data from two consecutive days in an example mouse. **c**, ROC curves when classifying the correlations in ISI histograms of the same versus different units within days (green) or matching versus nonmatching units across days (red). Data from two consecutive days in an example mouse. **d**, The AUC for many pairs of days, spaced by different amounts (red dots). The AUC of the same versus different units within recordings is also shown (green). Stability was estimated with a linear fit (curve). The discriminability remained highly stable, even across months. Data from an example mouse. **e**, AUC for many pairs of days, across many mice and recording locations. The $x$ axis indicates the number of

days between the pair. **f**, The correlation of the firing rate of a unit with other units forming a reference population that was tracked across days using UnitMatch. The correlation of a unit (black), one of its neighbors (top, blue) or its match on the next day (bottom, red) is shown. The neurons in the reference population were ordered by decreasing correlation with the unit from day 1 (black). **g–j**, The same as **b** (**g**), **c** (**h**), **d** (**i**) and **e** (**j**) but using the correlation with a reference population as a fingerprint. **k**, Comparison of the responses with natural images for a unit and one of its neighbors (top) or its match on the next day (bottom). The responses to the images were summarized by averaging over all images to obtain the time course (left) or averaging over time to obtain the responses to individual images (right). These two profiles were then concatenated to form a single fingerprint, which was compared across units. Images were ordered by decreasing response of the black unit. **l–o**, The same as **b** (**l**), **c** (**m**), **d** (**n**) and **e** (**o**) but using the responses to natural images as a fingerprint.

## Tracking over many recordings

So far, we have examined the matching of neurons across pairs of recordings, potentially spaced far apart. However, UnitMatch is easily scalable and can perform matching across all recorded days simultaneously, providing a match probability for all pairs of neurons in all recordings (Fig. 5a). The next stage is to use these probabilities to track individual neurons across multiple recordings, that is, to group matching units under a unique identification number (Fig. 1c).

UnitMatch provides a tracking algorithm that comes in three versions: default, liberal and conservative (Fig. 5b). The default version

of the algorithm iteratively inspects all pairs, and merges a unit with a target group of units if its probability of matching with all of the units in the target group that are within the recording and in neighboring recordings is higher than 0.5. This algorithm successfully tracked populations of neurons across days and weeks, allowing neurons to disappear and reappear across days (Fig. 5c,d). The more liberal version of the algorithm tracks more neurons at the cost of more false positives. Finally, the more conservative version ensures a higher probability of accurate tracking at the cost of more false negatives. These versions of the algorithm result in slightly different groupings (Extended Data

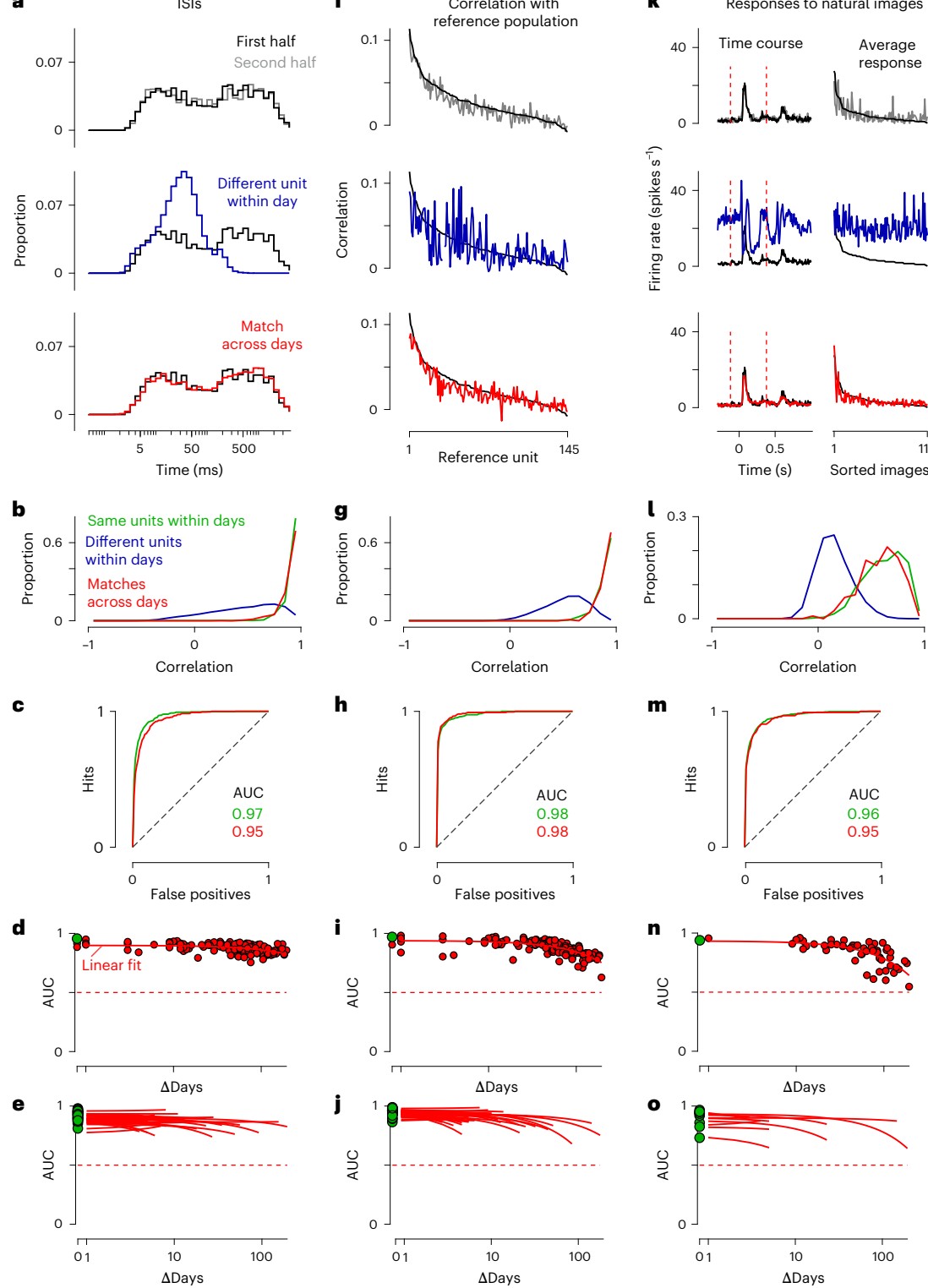

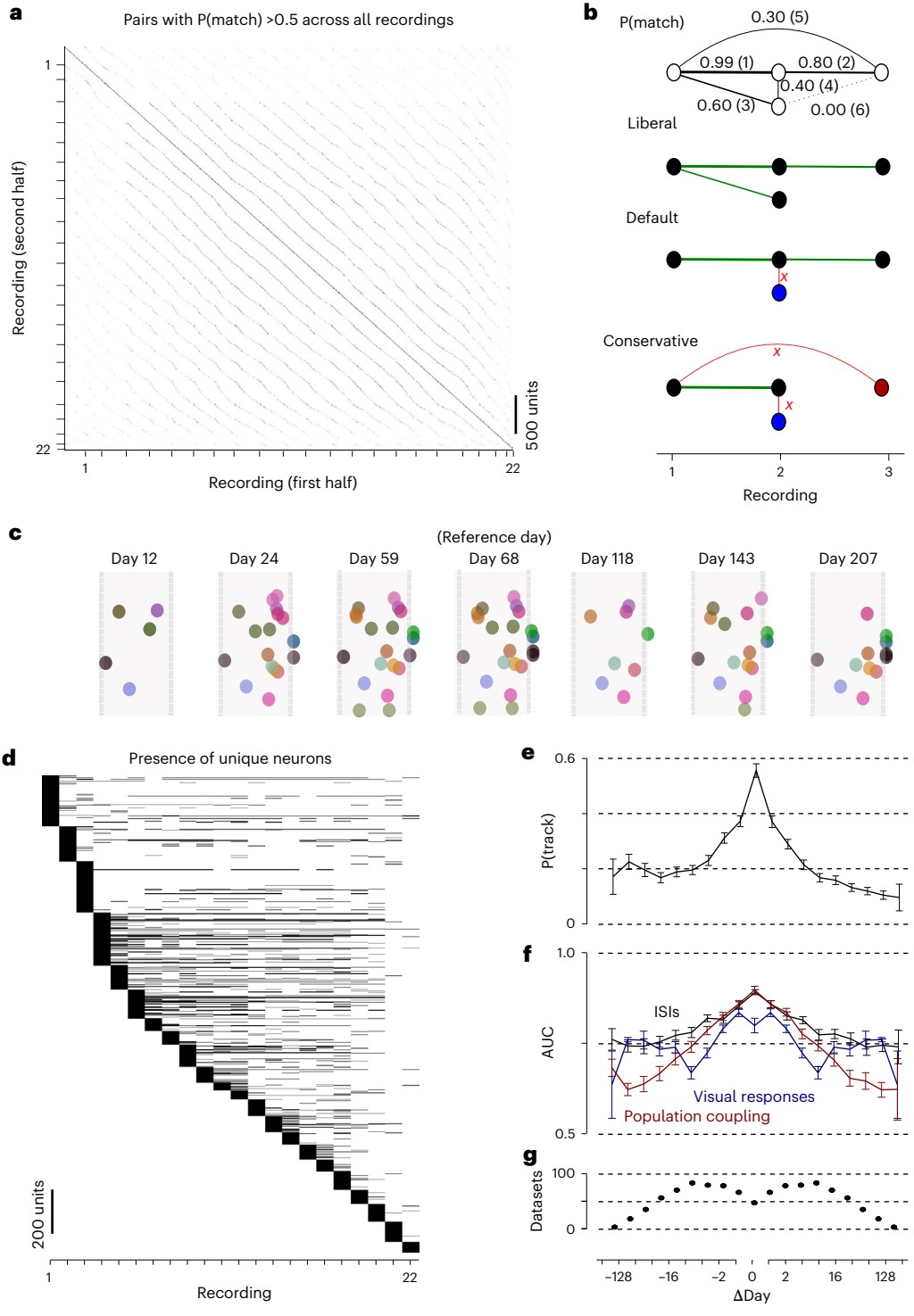

**Fig. 5 | Tracking neurons over many recordings. a**, The match probability P(match) of every pair of neurons in every pair of sessions, for an example mouse (ID1). The ticks on the axes indicate different recording sessions. **b**, Top: an illustration of unit tracking for four example units across three recordings, with match probabilities ranging from 0.00 to 0.99, sorted from high to low (brackets). Bottom: the outputs of the three versions of the algorithm to track neurons across many recordings, showing connections that support (green) or block (red) units being given the same identity (same color). For example, the 'default' version of the algorithm finds two distinct units, one appearing in all 3 days (black) and one appearing only on day 2 (blue). **c**, A population of neurons that was tracked over 195 days in the example mouse. Other neurons on the probe

are not shown. Note that some neurons can disappear on some days and reappear on other days. **d**, The presence (black) of a unique neuron across recordings, sorted by first appearance. **e**, The average probability of tracking a unit P(track) (±s.e.m.) as a function of days between recordings. The number of tracked neurons was divided by the total number of neurons available in the future recording (negative Δdays) or past recording (positive Δdays). The number of datasets per bin is indicated in **g**. **f**, The average AUC values (±s.e.m. across datasets) when comparing the functional similarity scores of tracked versus nontracked neurons, for ISI histogram correlations (black), correlation with reference population (red) and responses to natural images (blue). The number of datasets per bin is indicated in **g**. **g**, The number of datasets per bin.

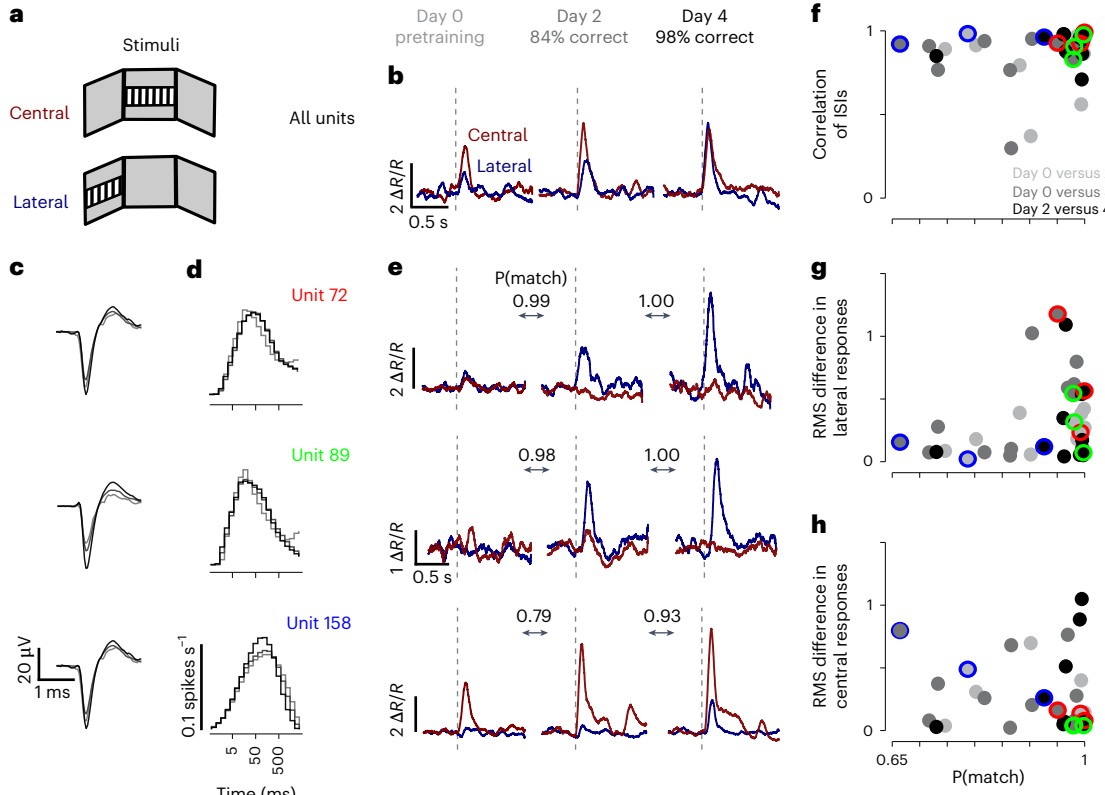

**Fig. 6 | Tracking units across learning. a**, Mice were implanted with chronic Neuropixels probes in the striatum and were trained to move a stimulus from the lateral screen to the central screen using a steering wheel. The responses of the neurons were then measured again during passive viewing of lateral and central stimuli. **b**, The average baseline (−0.2 to 0 s before stimulus onset) normalized firing rate (Δ*R/R*) aligned to stimulus onset (dashed line) during passive viewing of central stimuli (red) or lateral stimuli (blue), sorted by training day (left: day 0, middle: day 2, right: day 4) for the average population of tracked neurons (*n* = 12).

Behavioral accuracy of the mouse is expressed in percentage correct. **c,d**, The average waveforms (**c**) and ISI histograms (**d**) across 3 days for three example tracked neurons. **e**, The same as in **b**, but for individual neurons. **f**, The correlation of ISIs between days 0 and 2 (light gray), days 0 and 4 (gray) and days 2 and 4 (dark gray), for individual neurons (*n* = 12), against match probability P(match). The example neurons are indicated by color. **g**, The same as **f**, but for the root mean square difference between (normalized) lateral responses. **h**, The same as **f**, but for the root mean square difference between (normalized) central responses.

Fig. 7). As we will see below, the default algorithm is superior to the other two in some respects. We have thus used it for further analyses.

As might be expected, the probability of tracking neurons across multiple recordings decreased as the number of days between the recordings increased. To investigate the dynamics of tracking, we quantified the probability for a neuron recorded on a specific day to be tracked (not necessarily continuously) in the past (negative difference in days), or in the future (positive difference in days) (Fig. 5d,e). The probability of tracking within days (different recordings performed on the same day) was high but <1, setting an upper limit on the ability to track neurons in recordings spike-sorted independently. Interestingly, the tracking probability decreased in both directions, suggesting that neurons recorded by the probe are slowly but consistently renewed. However, the probability of a match was slightly lower in the future than in the past, indicating a progressive depletion of an initial pool of neurons. With this combination of implant and algorithms, up to 30% of neurons could be tracked for more than 100 days. These results, however, greatly depend on the quality and yield of the recordings.

Functional validation confirmed that the tracking algorithm performs accurately, even for recordings separated by many days (Fig. 5f). Indeed, the functional similarity scores for pairs identified as the same units by the algorithm remained higher than the ones of units labeled as 'different', across the whole spectrum of match probabilities (Extended Data Fig. 8a–f). When inspecting pairs of units between far-away recordings that had a match probability of 0, we observed that those

tracked by the algorithm had higher functional similarity scores than the pairs labeled as 'different'. Similarly, pairs of units that had a match probability of 1 but were identified as being different by the algorithm had a lower functional similarity score than those identified as tracked. This observation suggests that units were successfully tracked across recordings, beyond the simple match probabilities. This was especially true with the default version of the algorithm, which was thus superior to the liberal and conservative versions, with the best trade-off of false positives versus false negatives (Extended Data Fig. 8g–r).

### Tracking units across learning

We have shown that UnitMatch can be used to track units across days, and this can be validated by stable functional similarity scores. An important future application of this algorithm will be to track units as their functional properties evolve over time, particularly as a result of learning.

To illustrate this potential, we applied UnitMatch to a small, exploratory dataset recorded during a learning process. We trained a mouse (Extended Data Table 1) in a visuomotor operant task[47] and recorded activity in the dorsomedial striatum using a chronically implanted Neuropixels probe. The mouse was head-fixed in front of three screens with its forelimbs resting on a steering wheel. When the stimulus appeared on the left screen (contralateral to the recordings), moving the wheel clockwise moved the stimulus to the center, resulting in a sucrose water reward. The mouse learned to correctly move the wheel over a training period of a few days[47].

UnitMatch revealed intriguing changes in the activity of neurons across days. After each training day, we recorded passive responses to the presentation of a stimulus in the center screen or the left (contralateral) screen (Fig. 6a). We analyzed data recorded during passive viewing of the same set of stimuli on day 0 (pretraining), day 2 (intermediate performance) and day 4 (plateau performance). The population's response (averaged across all tracked neurons) to the central stimulus increased on day 2, and its response to the lateral stimulus increased on day 4 (Fig. 6b). However, tracking neurons with UnitMatch (for example, Fig. 6c–e) revealed substantial diversity across neurons. Some units increased their response to both stimuli over learning (for example, unit 158). Others developed a strong response to only one of the stimuli (for example, units 72 and 89). Despite changes in responses induced by learning, ISI histograms remained relatively stable (Fig. 6d). Importantly, there was no relation between match probability and changes in functional measures (Fig. 6f–h).

This proof of concept suggests that UnitMatch is a promising tool to reveal not only invariance but also plasticity in neural activity across days.

## Discussion

UnitMatch fills a need for flexible and probabilistic tracking of neurons across recordings, and has many advantages. First, it does not use functional properties in matching neurons, like many other algorithms do[6,7,9–12,16,19,22,28,29,39], allowing the user to ask whether functional properties change or remain constant. Second, it acts after spike sorting, allowing the user to choose the spike-sorting algorithm that they prefer. This is important because the quality of sorting algorithms keeps improving, and sorting is time-consuming and, thus, ideally done only once per recording. Because UnitMatch is based solely on the average waveform of neurons from single recordings, it is thus compatible with widely used preprocessing electrophysiology pipelines such as SpikeInterface[48], for which we provide example interfacing code. Third, it is specifically designed to handle long sequences of separate recordings, rather than the single prolonged recording required by other approaches[20,21,24]. Fourth, it uses within-recording cross-validation to build probability distributions and extract match probabilities. Consequently, it can also check for units that should have been merged or split within a single (potentially acute) recording. Fifth, unlike existing algorithms, it outputs match probabilities rather than a binary output, and provides a user interface for curation. Sixth, it is robust to the drift that is often observed in chronic Neuropixels recordings. Finally, it is substantially faster and performs better than even the latest algorithm in the field[46].

Although UnitMatch could track the same units over months, the number of units that were tracked decreased with time. This decrease could derive from numerous sources independent of the algorithm, such as a decline in recording quality, accumulation of drift across recordings, neurons becoming silent or dying, or changes in waveform properties. Indeed, the probability of finding a match depends on its quality metrics, suggesting an important role of recording quality and drift. Ideally, further work will reveal the contribution of each of these factors to the quality of the tracking.

UnitMatch revealed that distinctive functional properties of neurons remain remarkably stable over time; hence, it is tempting to use functional properties themselves to track neurons. However, this would prevent any evaluation of the variation in functional properties across time, and such a variation has been documented[4,5,19]. For example, the slow decrease in AUC values that we observed across days could be explained either by a decrease in the quality of matching or by changes in functional properties of the units. Therefore, unless there is reason to believe that the functional properties are constant[9,16,38], it is prudent to exclude these properties from the criteria that determine the tracking of units and only consider them as a possible validation[11,14,15,18,31] or as a separate question[6,7,22,23,25,38,39].

Taken together, these findings show that UnitMatch is a promising tool to characterize neural activity spanning a multitude of brain regions and time scales, such as memory, learning and aging.

## Online content

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

## Methods

Experimental procedures were conducted at University College London according to the UK Animals Scientific Procedures Act (1986) and under personal and project licenses released by the Home Office following appropriate ethics review.

We analyzed the data from 25 chronically implanted mice of both sexes with a Bl6 background. Mice were 3–9 months of age at implantation surgery and were implanted for maximally 8 months. During the experiments, mice were typically head fixed and exposed to sensory stimuli, engaged in a task, or resting. The mice were recorded from different experimental rigs, implanted and recorded by different experimenters using different devices (Extended Data Table 1).

### Surgeries

A brief (-1 h) initial surgery was performed under isoflurane (1–3% in $O_2$) anesthesia to implant either a titanium headplate (-25 × 3 × 0.5 mm, 0.2 g in the case of the Apollo implant) or a steel headplate (-25 × 5 × 1 mm, 0.5 g in the case of the ultralight and cemented implants). In brief, the dorsal surface of the skull was cleared of skin and periosteum. A thin layer of cyanoacrylate was applied to the skull and allowed to dry. Thin layers of ultraviolet (UV)-curing optical glue (Norland Optical Adhesives #81, Norland Products) were applied and cured until the exposed skull was covered. The head plate was attached to the skull over the interparietal bone with Super-Bond polymer. In one mouse (ID 2/19), a silver wire was implanted in the mouse's skull to ground the mouse during recordings.

After recovery, mice were treated with carprofen or meloxicam for 3 days, then acclimated to handling and head fixation. Mice were then implanted with either a modular recoverable[35], ultralight or cemented implant (see section 'Implants' below). Briefly, craniotomies were performed on the day of the implantation, under isoflurane (1–3% in $O_2$) anesthesia and after injection of Colvasone and Rimadyl. The UV glue was removed, and the skull was cleaned and scarred for best adhesion of the cement. The skull was leveled, before opening the craniotomies using a drill or a biopsy punch. Once exposed, the brain was covered with Dura-Gel (Cambridge Neurotech).

### Implants

**Cemented implant.** Four mice were implanted by holding and inserting the probes using a cemented dovetail and applying dental cement to encase the probe printed circuit board and reliably attach it to the skull. The recordings were made in external reference mode, using the silver wire or the headplate as the reference signal. The data from these four mice were already published[31].

**Recoverable modular implants.** Twenty mice were implanted with a recoverable, modular implant. The methods for the Apollo implant[35] and the 'Haesler' implant[31,34] have been described in their respective papers. The third implant ('Repix'[36] implant) is conceptually similar. In short, the implant was held using the three-dimensionally printed payload holder and positioned using a micromanipulator (Sensapex). After carefully positioning of the shanks at the surface of the brain, avoiding blood vessels, probes were inserted at slow speed (3–5 μm s⁻¹). Before surgery, the probes were coated with fluorescent dye DiI (ThermoFisher) by either manually brushing each probe with a droplet of DiI or dipping them in directly in DiI, for histological reconstruction. Once the desired depth was reached (optimally, just before the docking module touched the skull), the implant was sealed using UV glue, then covered with Super-Bond polymer, ensuring that only the docking module was cemented. After finishing all recording sessions, the probes were explanted and cleaned before reusing. The recordings were made in external or internal reference mode, using the headplate as the reference signal.

**Ultralight implant.** We also developed an ultralight implant (https://github.com/Julie-Fabre/ultralight_implant). Briefly, one Neuropixels probe was encased in rigid-resin K custom-made three-dimensionally printed parts. A thin square of sorbuthane sheet was added to the front of the implant. Special care was taken to ensure all shanks were parallel to each other and to the implant. This implant was then slowly lowered into the brain. At the target depth, the implant base was covered in Vaseline to protect the shank from subsequent cement applications. We then applied cement to the implant and mouse skull. To explant, we carefully drilled the implant out in the areas where Vaseline had been applied.

### Data processing

Electrophysiology data were acquired using SpikeGLX (https://billkarsh.github.io/SpikeGLX/), and each session was spike-sorted with Kilosort2.5[42] or Kilosort4[49] (only for Fig. 6). Data were preprocessed using 'ExtractKilosortData.m', meaning that all relevant information was extracted (for example, positions of recording sites, information on extracted clusters and their spike times) and common noise was removed. Well-isolated units were selected using Bombcell[43] (https://github.com/Julie-Fabre/bombcell; using parameters defined in bc_qualityParamValuesForUnitMatch.m). For each session the average waveform on every recording site for each unit was extracted, either through Bombcell or through Unitmatch's 'ExtractAndSaveAverageWaveforms.m'.

Input to the core of UnitMatch, which matches units purely on the basis of waveforms, was information on the clusters, at least (1) cluster identity, (2) a Boolean on which clusters to include, typically well-isolated units, (3) which recording session it was recorded in, and (4) on which probe it was recorded. In addition, it requires parameters (we used default parameters available using 'DefaultParametersUnitMatch.m') containing information on where to find the raw waveforms.

Example analysis pipelines from raw electrophysiology recorded using SpikeGLX all the way to using and validating UnitMatch are provided in the UnitMatch repository. A minimal use case scenario is also provided in 'DEMO_UNITMATCH.m', which is also useful for electrophysiological data recorded and preprocessed using other probes and software.

### Mathematical definitions

We consider recordings made in a probe with $N$ sites, and we denote with $\mathbf{p}_s$ the position of site $s$ (a vector with the $x$, $y$ coordinates). For every unit $i$, we denote the spike waveform at site $s$ and at time $t$ as $w_{s,t,i}$ (averaged across $n$ spikes of that neuron).

**Step 1: extract waveform parameters.** Some useful summaries of the spike waveform include the spatial footprint

$$w_{s,i}^* = \max_t(|w_{s,t,i}|) \tag{1}$$

and the maximum site $s_i^*$ where the voltage has maximum amplitude

$$s_i^* = \arg\max_s(w_{s,i}^*). \tag{2}$$

Most analyses are performed in a time window of size $T$ samples starting 0.23 ms before the waveform reaches its peak and ending 0.50 ms after the peak. To establish a baseline noise level, we used a window of same duration starting 1.33 ms before waveform onset.

The spatial decay of the waveform is the degree to which the waveform's maximum amplitude at site $s$ decreases as a function of distance from the peak site, $|\mathbf{p}_s - \mathbf{p}_{s_i^*}|$. To describe it, we fit an exponential decay function (Fig. 2c) with scale $\lambda_i$ such that

$$w_{s,i}^* \approx A_i \exp\left(-\lambda_i |\mathbf{p}_s - \mathbf{p}_{s_i^*}|\right), \tag{3}$$

and we use this fit to obtain the distance at which the amplitude drops to 10% of maximal value, $d_{10} = \log(10)/\lambda$. For further analysis, we only take recording sites into account with distance to $s_i^* < d_{10}$.

The centroid trajectory of neuron $i$ is (Fig. 2f)

$$\mathbf{c}_{t,i} = \frac{\sum_s w_{s,t,i} \mathbf{p}_s}{\sum_s w_{s,t,i}}, \tag{4}$$

and its travel direction at each time $t$ is

$$\theta_{t,i} = \tan^{-1} \frac{|x_{t,i} - x_{t+1,i}|}{|y_{t,i} - y_{t+1,i}|}, \tag{5}$$

$x_{t,i}$ and $y_{t,i}$ being the components of $\mathbf{c}_{t,i}$.

The neuron's average centroid (Fig. 2f) is

$$\mathbf{c}_i^* = \frac{\sum_s w_{s,i}^* \mathbf{p}_s}{\sum_s w_{s,i}^*}. \tag{6}$$

To calculate a neuron's average waveform, we start by computing the proximity $f_{s,i}$ of each site $s$ to the centroid of the neuron $\mathbf{c}_i^*$

$$f_{s,i} = 1 - \frac{|\mathbf{p}_s - \mathbf{c}_i^*|}{d_{10}}, \tag{7}$$

where $d_{10}$ is the distance where amplitude drops to 10% (or 150 μm if that distance is larger). At sites that are further away (where $f_{s,i}$ would be negative), we set $f_{s,i} = 0$.

We then calculate the unit's spatial decay as the average decrease in amplitude divided by the increase in distance for all sites closer than $d_{10}$ (Fig. 2):

$$d_i = \frac{1}{N} \sum_{s \neq s_i^*} \frac{w_{s_i^*,i}^* - w_{s,i}^*}{|\mathbf{p}_{s_i^*} - \mathbf{p}_s|}. \tag{8}$$

We then compute the neuron's weighted-average waveform $\overline{w}_{t,i}$ (Fig. 2e) as

$$\overline{w}_{t,i} = \frac{\sum_s f_{s,i} w_{s,t,i}}{\sum_s f_{s,i}}. \tag{9}$$

We use this waveform to compute the weighted amplitude of the neuron's spike as

$$a_i = \max_t (|\overline{w}_{t,i}|). \tag{10}$$

When comparing waveforms between units, we normalize $\overline{w}$ to obtain

$$\hat{w}_{t,i} = \frac{\overline{w}_{t,i} - \min_t \overline{w}_{t,i}}{\max_t \overline{w}_{t,i} - \min_t \overline{w}_{t,i}}. \tag{11}$$

**Step 2: compute similarity scores.** Based on these parameters, we next compute similarity scores for each pair of units $i$ and $j$. These scores are scaled between 0 and 1, with 1 being the most similar. For most similarity scores, we do '0–99 scaling': we rescale the similarity scores so that the minimum is 0 and the 99th percentile is 1. If $X_{i,j}$ is the similarity score between units $i$ and $j$, its 0–99 scaling is

$$|X_{i,j}|_{99} = \frac{P_{99}(X) - X_{i,j}}{P_{99}(X) - P_0(X)}, \tag{12}$$

where $P_K(X)$ is the $K$th percentile of $X$. For similarity scores above the 99th percentile, we clip the score to 1.

We used two types of similarity score: those based on waveform time courses and those based on waveform trajectories.

*Amplitude similarity.* We compute the difference in maximum amplitude between each unit $i$ and $j$, and we apply 0–99 scaling to its square root via

$$A_{i,j} = \left| \sqrt{|a_i - a_j|} \right|_{99}. \tag{13}$$

*Decay similarity.* We compute the difference in spatial decay, and we apply 0–99 scaling to it via

$$D_{i,j} = \|d_i - d_j\|_{99}. \tag{14}$$

*Waveform similarity.* We compute the Euclidean distance between the waveforms, and we apply 0–99 scaling to it via

$$E_{i,j} = \left| \left\langle \left( \hat{w}_{t,i} - \hat{w}_{t,j} \right)^2 \right\rangle_t^{1/2} \right|_{99}. \tag{15}$$

We also compute the correlation between the waveforms and apply Fisher's $z$ transformation and 0–99 scaling to it via

$$v_{t,i} = \overline{w}_{t,i} - \langle \overline{w}_{t,i} \rangle_t$$

$$\rho_{i,j} = \frac{\langle v_{t,i} v_{t,j} \rangle_t}{\sqrt{\langle v_{t,i}^2 \rangle_t \langle v_{t,j}^2 \rangle_t}} \tag{16}$$

$$Z_{i,j} = \left| 1 - \log \left( \frac{1 + \rho_{i,j}}{1 - \rho_{i,j}} \right) \right|_{99}. \tag{17}$$

Empirically, we found both measures (distance and correlation) to be informative. Of course, they are also highly correlated with each other (Extended Data Fig. 9b). This correlation poses problems for a naive Bayes decoder. To take them both into consideration, we defined 'waveform similarity' as their average:

$$W_{i,j} = (E_{i,j} + Z_{i,j})/2. \tag{18}$$

*Centroid similarity.* We compute the mean absolute distance between two centroids, and then we rescale it to obtain a measure of proximity that is 1 if centroids are identical and 0 if they are further than $d_{max} = 100$ μm:

$$d_{i,j} = \left\langle |\mathbf{c}_{t,i} - \mathbf{c}_{t,j}| \right\rangle_t \tag{19}$$

$$C_{i,j} = \left[ \frac{d_{max} - d_{i,j}}{d_{max} - \min_{ij} d} \right]_+. \tag{20}$$

Units that are further apart than $d_{max}$ are unlikely to be a match, even when considering drift between recordings.

*Volatility similarity.* If some of the drift remains uncorrected, a unit that appears in two recordings may have centroid trajectories that are identical but displaced by a constant shift. To correct for this, we subtracted the average centroid (equation (6)) from the centroid trajectory (equation (4)) for each unit and computed their similarity $F_{i,j}$ across units as in equations (19) and (20):

$$F_{i,j} = \left| \left\langle |(\mathbf{c}_{t,i} - \mathbf{c}_i^*) - (\mathbf{c}_{t,j} - \mathbf{c}_j^*)| \right\rangle_t \right|_{99}. \tag{21}$$

We also compute the standard deviation in Euclidean distance between centroids, and apply 0–99 scaling to it via

$$S_{i,j} = \left| \left\langle \left( |\mathbf{c}_{t,i} - \mathbf{c}_{t,j}| - d_{i,j} \right)^2 \right\rangle_t^{1/2} \right|_{99}. \tag{22}$$

Since $F_{i,j}$ and $S_{i,j}$ are highly correlated (Extended Data Fig. 9b), we averaged these two scores to centroid 'volatility' similarity via

$$V_{i,j} = (F_{i,j} + S_{i,j})/2. \tag{23}$$

*Route similarity.* We compute the summed difference in direction (angle) of the centroid trajectory, and apply 0–99 scaling to it via

$$\Theta_{i,j} = \left| \langle \theta_{t,i} - \theta_{t,j} \rangle_t \right|_{99}. \tag{24}$$

In addition, we compute the distance traveled by the centroid between each time point of the trajectory and compare the differences between each pair of units $i$ and $j$, and apply 0–99 scaling via

$$P_{i,j} = \left| \sqrt{\sum_{t \in T} ||\mathbf{c}_{t+1,i} - \mathbf{c}_{t,i}|| - |\mathbf{c}_{t+1,j} - \mathbf{c}_{t,j}||} \right|_{99}. \tag{25}$$

The final route similarity is

$$R_{i,j} = (\Theta_{i,j} + P_{i,j})/2. \tag{26}$$

*Default similarity scores.* Before settling on this set of default similarity scores, we evaluated the performance of other scores (Extended Data Fig. 9). For each set of scores, we computed the AUC value in classifying whether two waveforms came from the same unit or not (Extended Data Fig. 9a). This process led us to consolidate similarity scores that were highly correlated with each other (Extended Data Fig. 9b). Note that, based on within-day cross-validated performance, a user of UnitMatch will be advised what similarity scores to use for every individual dataset. In this paper, we only used default parameters and scores.

**Step 3: identify putative matches.** Having defined these six similarity scores for each pair of units $i$ and $j$, we averaged them to obtain a total score

$$T_{i,j} = (A_{i,j} + V_{i,j} + W_{i,j} + C_{i,j} + D_{i,j} + R_{i,j})/6. \tag{27}$$

We define the preliminary class ($M$) of a pair of units as

$$\begin{cases} M = 1 \text{ if } T_{i,j} > T_{P(M_{i,j}=0)} \\ \quad\quad\quad \text{else } M = 0 \end{cases}, \tag{28}$$

where $T_{P(M_{i,j}=0)}$ is defined as the crossing point of probability distributions of $T_{i,i}$ and $T_{i,j}$, with $j$ within 50 μm of $i$. In the case of overall lower scores across days (for example, due to uncorrected drift), we lowered the threshold by the difference in means (by fitting a normal distribution) for the within-day distribution (Fig. 3h, blue and red green curves combined) and the across-days distribution (Fig. 3i, red curves).

**Step 5: build classifier.** We use the preliminary class labels to build the probability distributions for the similarity scores as defined above, and use these to compute the probability of a match between units $i$ and $j$ as

$$P(M_{i,j} = 1 | \mathbf{X}_{i,j}) = \frac{P(M_{i,j} = 1) \prod_{p=1}^{n} P(X_{i,j,p} | M_{i,j} = 1)}{\sum_k P(M_{i,j} = k) \prod_{p=1}^{n} P(X_{i,j,p} | M_{i,j} = k)}, \tag{29}$$

where $\mathbf{X}_{i,j}$ is the vector of elements $X_{i,j,p}$ containing the individual similarity scores ($A_{i,j}$, $V_{i,j}$ and so on).

**Functional similarity scores**
To evaluate UnitMatch performance, we determined three functional similarity scores of neuronal activity.

**ISI fingerprint.** For each neuron $i$ we compute the ISI histogram $\mathbf{A}_i$ of elements $a_{i,\tau}$ as the distribution of the times between consecutive spikes, binned on a logarithmic scale from 0 to 5 s. The ISI histogram $\mathbf{A}_i$ was then use as the first functional fingerprint.

**Cross-correlation fingerprint.** We computed the correlation of each unit with a reference population of units that was tracked across days. For each day $d$, we first binned the spiking activity of each unit across each half of the session using bins of 10 ms. Then, we computed the cross-correlation of each unit with every unit that was found to be tracked across days, yielding vectors $\mathbf{C}_i$ of elements $c_{i,j}$ corresponding to the instantaneous correlation coefficient of unit $i$ with unit $j$. The value of the correlation of one unit with itself if the unit was part of the reference population was set to NaN. These vectors $\mathbf{C}_i$ were used as the second functional fingerprint.

**Natural image responses fingerprint.** To characterize the functional fingerprint of the neurons in visual cortex, we showed 112 natural images, each presented five times in a random order, to the head-fixed mice[31]. Two versions of the protocol were used, one long (1 s stimulus, 2 s intertrial interval) and one short (0.5 s, 0.8 s), without affecting the overall reliability of the fingerprints. To define the fingerprint, we computed the responses as the peristimulus histograms locked on the image onset (0.3 s before and 0.5 s after) and the image offset (from 0 to 0.5 s after), using 5 ms bins. The response $R_{i,t,s}$ for each unit $i$ and stimulus $s$ were then defined as the concatenation of the onset and offset matrices along their temporal dimensions. Finally, two fingerprints were obtained by looking both at the average time course

$$p_{i,t} = \langle R_{i,t,s} \rangle_s \tag{30}$$

and the average response to each image

$$m_{i,s} = \langle R_{i,t,s} \rangle_t. \tag{31}$$

We then concatenated the vectors of elements $p_{i,t}$ and $m_{i,s}$ for each unit $i$ to obtain its third functional fingerprint.

**Fingerprint stability.** To assess the similarity $S_{i,j,d_1,d_2}$ of the fingerprints of the units $i$ and $j$ across two days $d_1$ and $d_2$, we first computed the fingerprints separately for both halves of the recording sessions, yielding two fingerprints $f_{i,d_1,1}$ and $f_{i,d_1,2}$ for each unit. Then, we computed the correlation of the fingerprint of units $i$ and $j$ across the two days and using different halves via

$$S_{i,j,d_1,d_2} = \mathrm{corr}(f_{i,d_1,1}, f_{j,d_2,2}). \tag{32}$$

Using two different halves allowed use to compute the fingerprint's reliability when $d_1 = d_2$.

**ROC and AUC.** To quantify the amount of information present in the distributions of the correlations of the fingerprints, we computed the ROC curve for different populations of pairs: pairs coming from the same units, or different units within days, or pairs coming from putative matched units, or nonmatched units, across days. We then computed the area under the ROC curve (AUC) to quantify this difference between distributions.

Only sessions with at least 20 matched units were taken into consideration. Moreover, in the case of the natural image responses fingerprint, these units had to be reliable on the first day (test–retest reliability of the fingerprint >0.2). Units that had a match within recordings were excluded from this analysis. For each mouse, the AUCs were then averaged across recordings locations. Similarly, the slope of AUC versus days was computed for each recording location (whenever there

were at least 3 days recorded at that location), and all slopes for each mouse were then averaged. Statistics were performed across animals.

### Continuous tracking algorithms

To track neurons across many recordings, we developed three versions of an algorithm. They all rely on the same procedure of serially going through all pairs of units but have different rules to group units under a common identification number. First, all the pairs (across all recordings) are sorted by their probability of matching. Then, the three versions will consider attributing the same unique identification number to the two members of the pair (and to all members of their respective groups) if they have a probability of matching that is above 0.5. The liberal version has no other constraint. The conservative version, on the other hand, will group these units only if all members of both groups match with each other. The intermediate version, finally, does something in between: it will group these units if each unit of the pair matches with all the units from the other group that are either in the same or an immediately adjacent recording.

To compute the probability of a unit being tracked, we then looked at each unit across all recordings and computed the probability of this unit being tracked in previous or subsequent recordings. These probabilities were then averaged across all the units from each animal, and averaged across animals. AUCs were computed similarly to previously described.

### Tracking functional changes with learning

To track neurons across learning, we sorted data from three days with Kilosort4. We found 12 neurons tracked across the three recording days and computed the average baseline-corrected response to stimuli presented on the (contra)lateral and central screen. To do so, we computed the firing rate $R_{i,d,t}$ of unit $i$ on day $d$ around the stimulus time, averaged across trials, and normalized it to obtain the unit's response

$$\bar{R}_{i,d,t} = \frac{R_{i,d,t} - \langle R_{i,d,t}\rangle_{t<0}}{\langle R_{i,d,t}\rangle_{t<0}},\tag{33}$$

where $\langle R_{i,d,t}\rangle_{t<0}$ denotes the baseline firing rate of the unit. It was then smoothed with a moving average window for plotting.

To evaluate the stability of functional measures, we quantified the Pearson correlation between ISI histograms across days, and the root mean square $RMS_i$ of the normalized visual responses across days via

$$RMS_i = \sum_t \left(\bar{R}_{i,1,t} - \bar{R}_{i,2,t}\right)^2.\tag{34}$$

### Reporting summary

Further information on research design is available in the Nature Portfolio Reporting Summary linked to this article.

### Data availability

Example data for mouse IDs 1–5 (Extended Data Table 1) are available via figshare (https://doi.org/10.6084/m9.figshare.24305758.v1)[51] as part of the software demo. Full datasets for mice 1, 7 and 8 are available via figshare (https://doi.org/10.5522/04/24411841.v1)[52], and due to size constraint, the rest of the full datasets can only be made available upon request.

### Code availability

UnitMatch software is available in Matlab and Python via GitHub at https://github.com/EnnyvanBeest/UnitMatch or via Zenodo at https://zenodo.org/records/12734237 (ref. 50). The Python version includes a SpikeInterface plugin. UnitMatch is licensed under a Creative Commons Attribution-NonCommercial-ShareAlike 4.0 International

License. Bombcell is available in Matlab via Github at https://github.com/Julie-Fabre/bombcell or via Zenodo at https://doi.org/10.5281/zenodo.8172822 (ref. 43). Bombcell is under the open-source copyleft GNU General Public License 3.

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

## Acknowledgements

We thank M. Robacha for help with experiments and histology, B. Terry for animal husbandry and C. B. Reddy for lab management. This project received funding from the Wellcome Trust (Investigator Award 223144 to M.C. and K.D.H., Early Career Award 227065 to C.B.), NIH (U19NS123716), Biotechnology and Biological Sciences Research Council (grant BB/T016639/1 to M.C. and P.C.), the European Union's Horizon 2020 research and innovation program (Marie Skłodowska-Curie grant agreement no. 101022757 to E.H.v.B.) and European Molecular Biology Organization (ALTF 740-2019 to C.B.). M.C. holds the GlaxoSmithKline/Fight for Sight Chair in Visual Neuroscience.

## Author contributions

Conceptualization: C.B. and E.H.v.B.; methodology: C.B., M.C., K.D.H. and E.H.v.B.; software: C.B., S.W.D. and E.H.v.B.; formal analysis: C.B., J.M.J.F. and E.H.v.B.; investigation: C.B., J.M.J.F. and E.H.v.B.; resources: C.B., P.C., J.M.J.F., A.L., F.T. and E.H.v.B.; data curation: C.B., P.C., J.M.J.F., A.L., F.T. and E.H.v.B.; writing—original draft: C.B., M.C. and E.H.v.B.; writing—review and editing: C.B., M.C., P.C., S.W.D., J.M.J.F., F.T. and E.H.v.B.; visualization: C.B., M.C., J.M.J.F. and E.H.v.B.; supervision: M.C. and K.D.H.; funding acquisition: C.B., M.C., P.C., K.D.H. and E.H.v.B.

## Competing interests

The authors declare no competing interests.

## Additional information

**Extended data** is available for this paper at https://doi.org/10.1038/s41592-024-02440-1.

**Correspondence and requests for materials** should be addressed to Enny H. van Beest or Célian Bimbard.

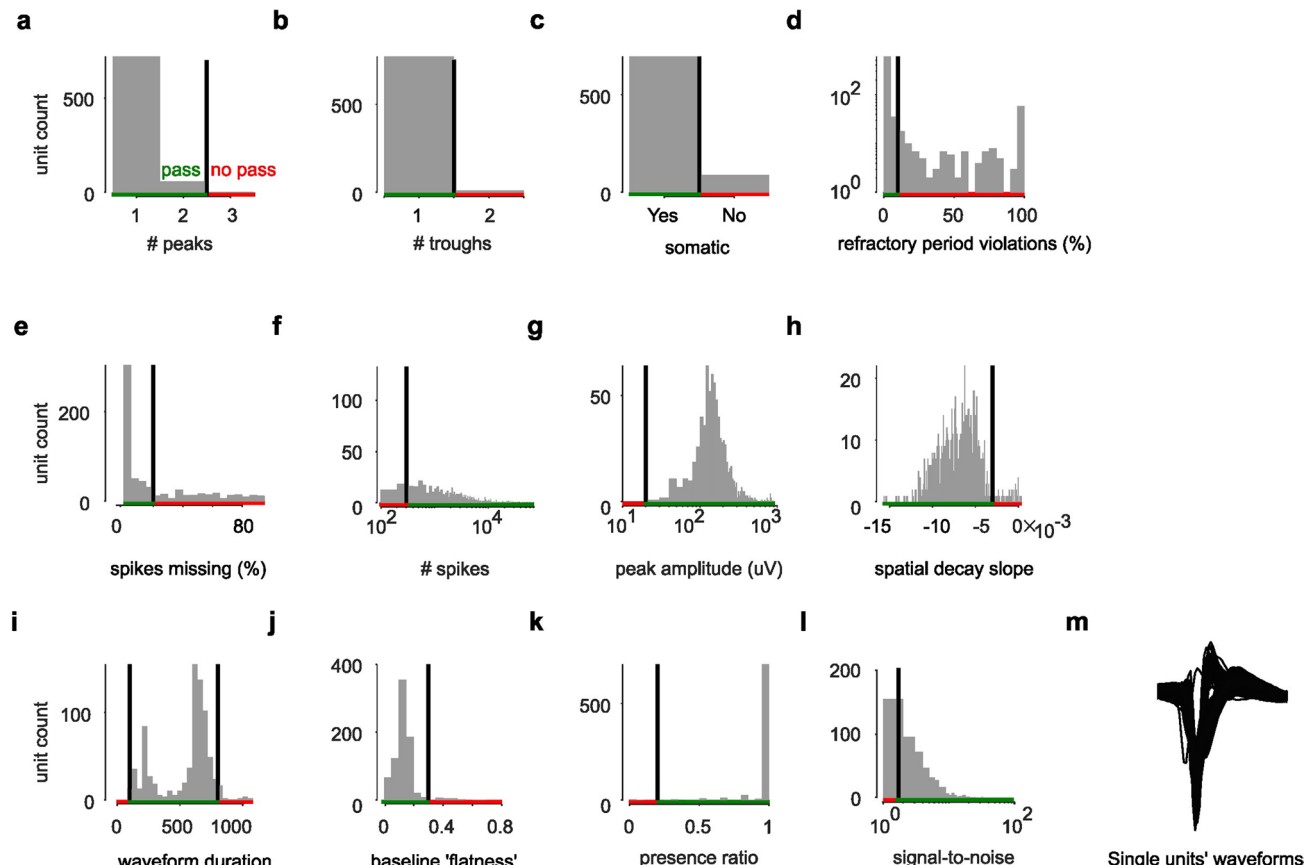

**Extended Data Fig. 1 | Bombcell output distributions for an example dataset.**
**a-l**, For all the units in an example dataset (Mouse ID 1, Extended Data Table 1) we measured 12 parameters. Each panel shows the number of units that passed (*green* section of the abscissa) or did not pass (*red* section) the selection based on that parameter. The parameters are: **a**, Number of detected peaks. **b**, Number of detected troughs. **c**, Somatic waveform. **d**, Estimated percentage of refractory period violations. **e**, Estimated percentage of spikes below the spike sorting algorithm's detection threshold, assuming a Gaussian distribution of spike amplitudes. **f**, Total number of spikes. **g**, Mean raw absolute waveform amplitude (μV). **h**, Spatial decay slope (fit). **i**, Waveform duration (μs). **j**, Waveform baseline 'flatness', defined as the ratio between the maximum value in the waveform baseline and the maximum value in the waveform. **k**, Presence ratio (of total recording time), defined as the fraction of bins that contain at least one spike. **l**, Signal-to-noise ratio. **m**, Waveforms of units that survived all quality metrics thresholds.

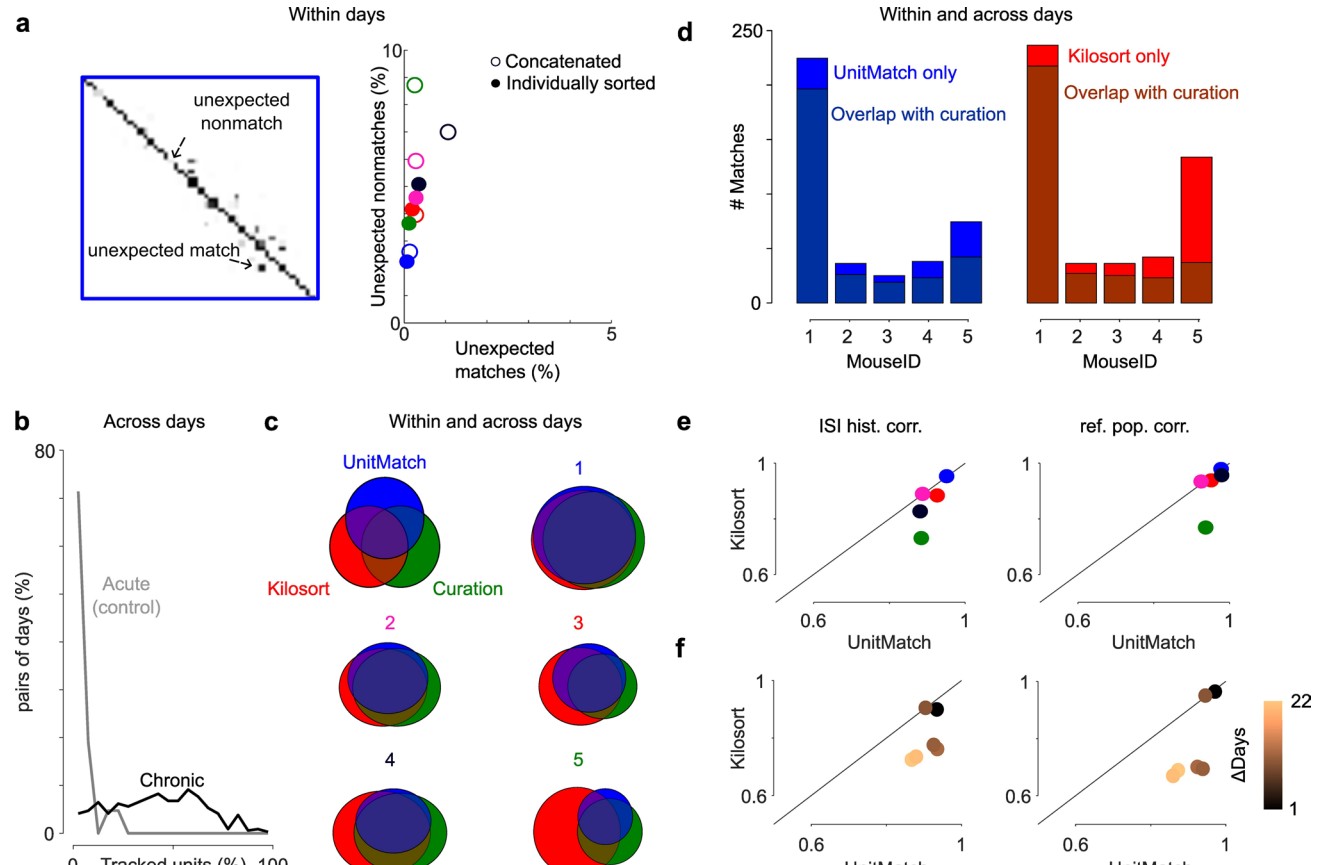

**Extended Data Fig. 2 | UnitMatch performance. a**, *Left:* Zoom in on Fig. 2j on some units along the main diagonal. On the diagonal we expect the matching probability P(match) to be close to 1, off diagonal we expect P(match) to be close to 0. *Right:* Unexpected matches (%) and nonmatches (%) by UnitMatch relative to units that were defined as good single units within a day by Kilosort. UnitMatch was either run on individually sorted data (*closed circles*), or on concatenated data (*open circles*). Colors depict individual mice, and refer to colors in **c. b**, Pairs of consecutive recording days (%) for acute (gray) and chronic (black) recordings as a function of the percentage of tracked units. The percentage of tracked units is defined as the number of matched units between two consecutive recording

days divided by the number of units on the recording day with the least recorded units. **c**, Venn diagrams for five individual mice illustrating the overlapping pairs of units assigned as 'match'. **d**, Units matched within or across two (concatenated) consecutive days by UnitMatch (left) and Kilosort (right) for five mice. Dark parts of the bars show the overlap with curated matches, light parts of the bar the matches additionally made by the respective algorithm. **e**, AUC value for ISI histogram correlations (left) and reference population correlations (right) for matches versus non-matches found by UnitMatch (x-axis) versus Kilosort (y-axis) for the same five mice as in C. **f**, same as **e**, but for multiple pairs of days of mouse ID1. ΔDays given by color bar.

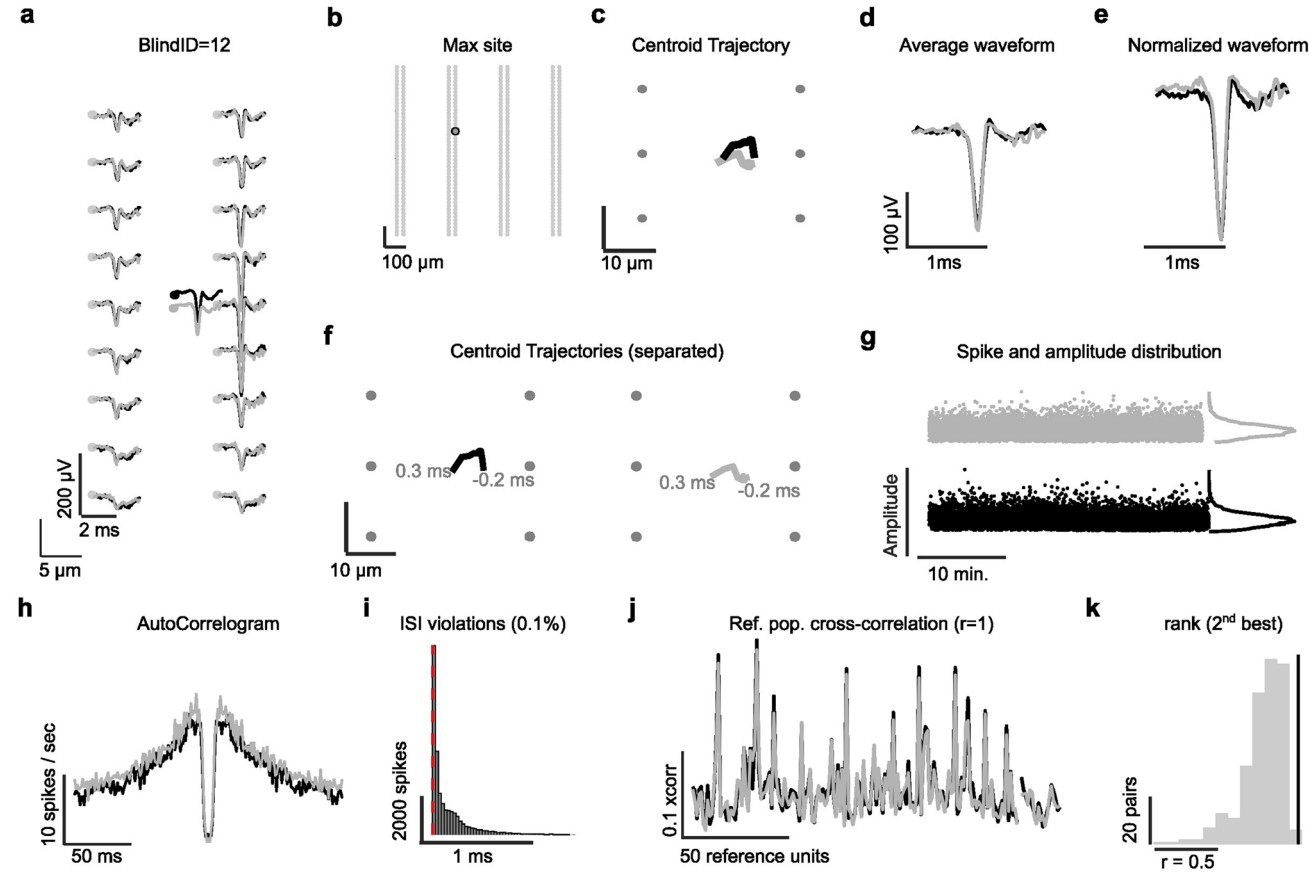

**Extended Data Fig. 3 | Expert curation.** Example of a figure seen by the six expert curators that were asked to judge whether the *black* and *gray* waveforms came from the same unit. Curators could score a pair with '1' (match), '0' (unsure), or '−1' (not a match). This example was found to be a match by both UnitMatch and stitched Kilosort. **a**, Average waveform across recording sites. **b**, Maximum recording site indicated (note, they overlap) on a 4-shank Neuropixels. **c**, Centroid trajectories. **d**, Average waveforms. **e**, Normalized waveforms (peak to base stretching). **f**, Same as **c**, but shown next to each other for black and gray unit. **g**, Spike times (x-axis) versus amplitude (y-axis), with the amplitude distribution next to it. Note that the amplitudes for 'gray' are drawn above 'black' for visibility. **h**, Autocorrelogram. **i**, Inter spike interval distribution. **j**, Reference population cross-correlation, which was 1 for this specific pair. **k**, reference population cross-correlation values between this pair of units (black line), relative to other possible pairs of units (distribution). Rank 2 means this cross-correlation value is the second highest of all possible pairs.

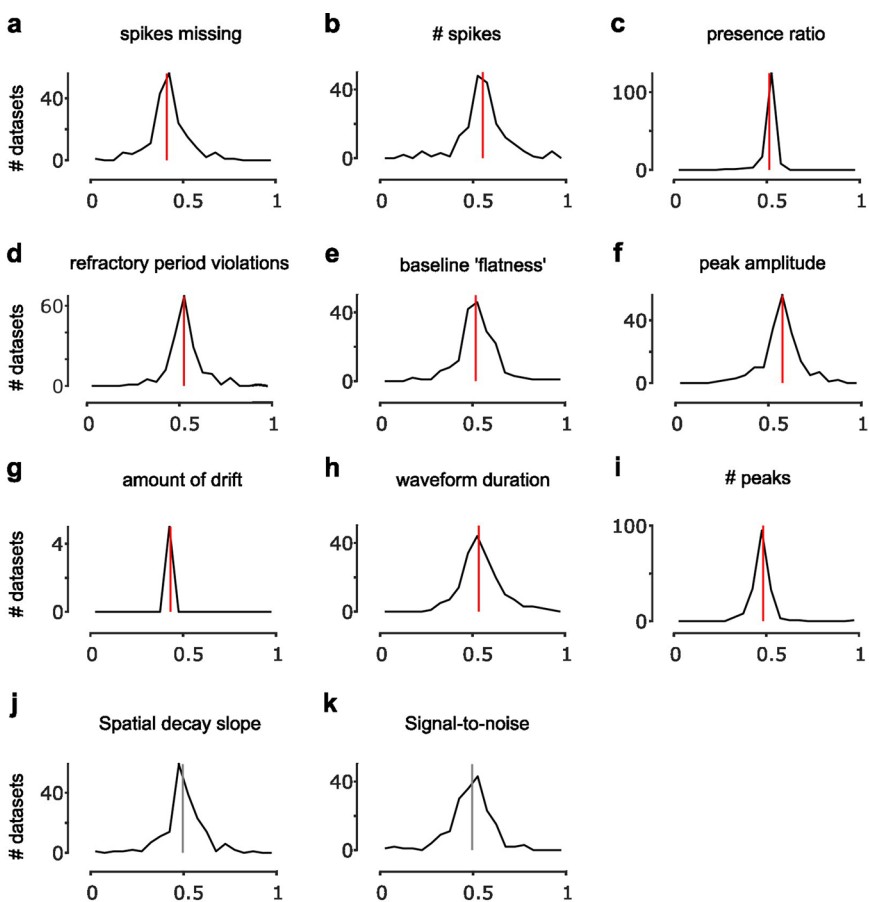

**Extended Data Fig. 4 | Quality metrics predict whether UnitMatch can find a match for a unit.** We evaluated the predictive value of different quality measures (Bombcell output; Fabre et al., 2023) on whether a match could be found for units included in our analysis. An AUC can be computed for each quality metric to quantify whether this metric differs across matches and non-matches. If the AUC is above (or below) 0.5, it means that units for which UnitMatch could find a match had a higher (or a lower) value for this parameter than the units that were left matchless. Each plot shows the distribution of AUC values and the median AUC value, across all 189 unique chronic recording datasets in 25 mice. To assess significance, we used two-sided t-tests (uncorrected for multiple comparisons). Red color of the median line indicates p<0.01. **a**, spikes missing, **b**, number of spikes, **c**, presence ratio, **d**, number of refractory period violations, **e**, baseline 'flatness', **f**, peak amplitude, **g**, amount of drift, **h**, waveform duration, **i**, number of peaks, **j**, spatial decay slope, and **k**, signal-to-noise ratio.

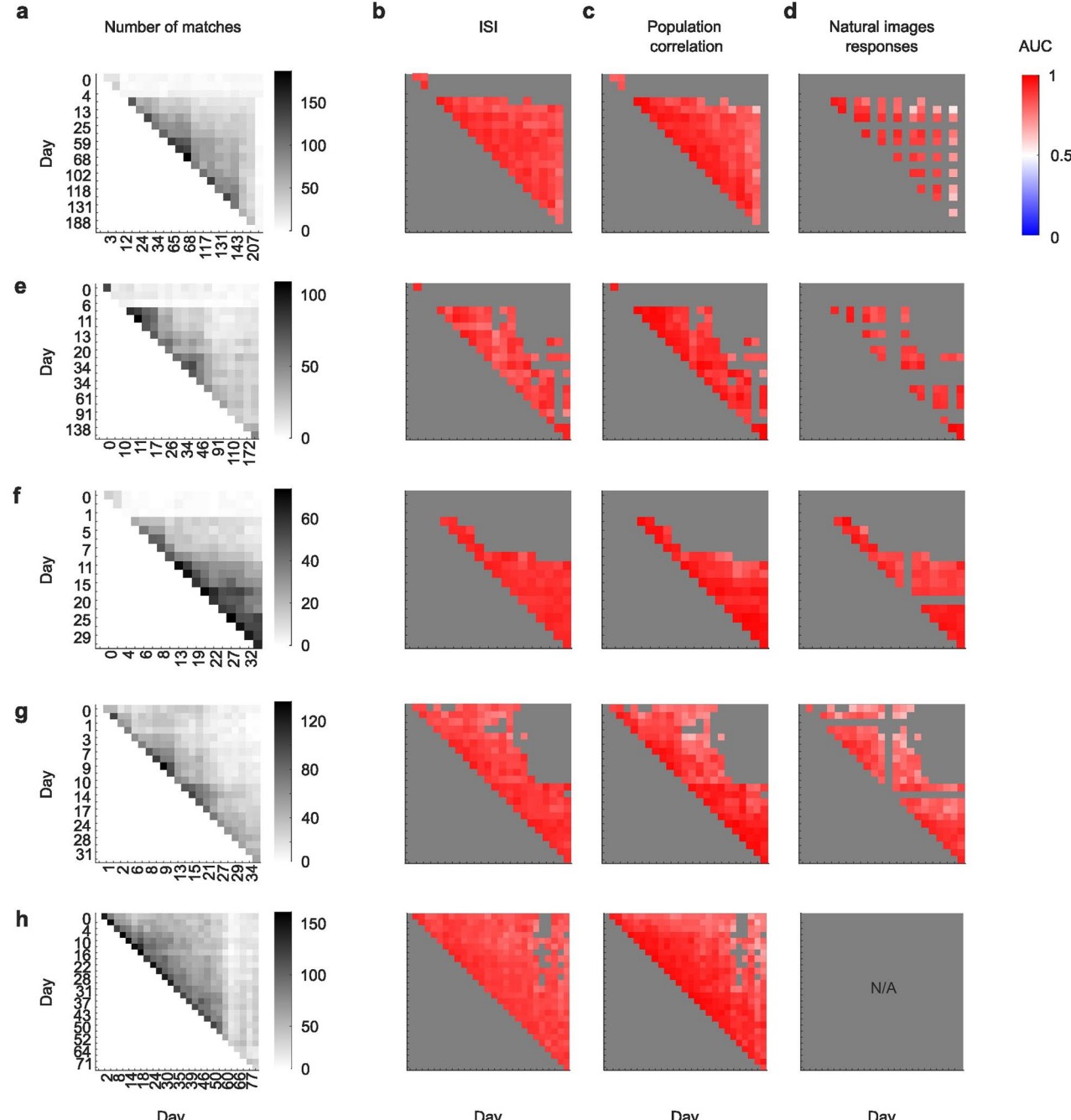

**Extended Data Fig. 5 | Units can be tracked across pairs of recordings separated by many days. a**, Number of matched units for each pair of days in one example animal. **b**, AUC of the inter-spike intervals (ISI). Only pairs of recordings with at least 20 tracked units are shown. **c**, AUC of the correlations with a reference population. **d**, AUC of the natural images responses. Protocols with natural images were not performed every day, and only recordings with at least 20 matched units are shown. **e-h**, Other example animals. All first four recordings were performed in visual cortex, and the last one was performed in frontal cortex.

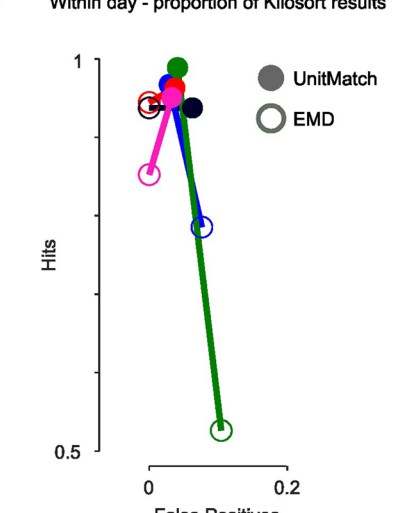

**a** Within day - proportion of Kilosort results

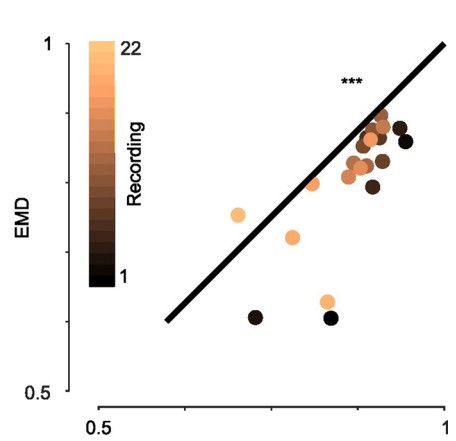

**b** AUC values across 22 recordings

**Extended Data Fig. 6 | Comparison to the EMD method. a,** We tested the performance of the EMD algorithm and UnitMatch on matching neurons from two halves of the same session in five mice. Note that this is a proportion relative to Kilosort output, and a small fraction of units from Kilosort should potentially have been merged. **b,** We compared matching performance in 22 recordings of example mouse 1. Since inter-spike-interval (ISI) histograms are generally stable (Fig. 4), we used the area under the curve (AUC) of ISI correlations to validate the pairs found by both algorithms. AUC values were significantly (***, paired t-test; t(20)=4.57, p=0.0002) larger for UnitMatch than for the EMD algorithm.

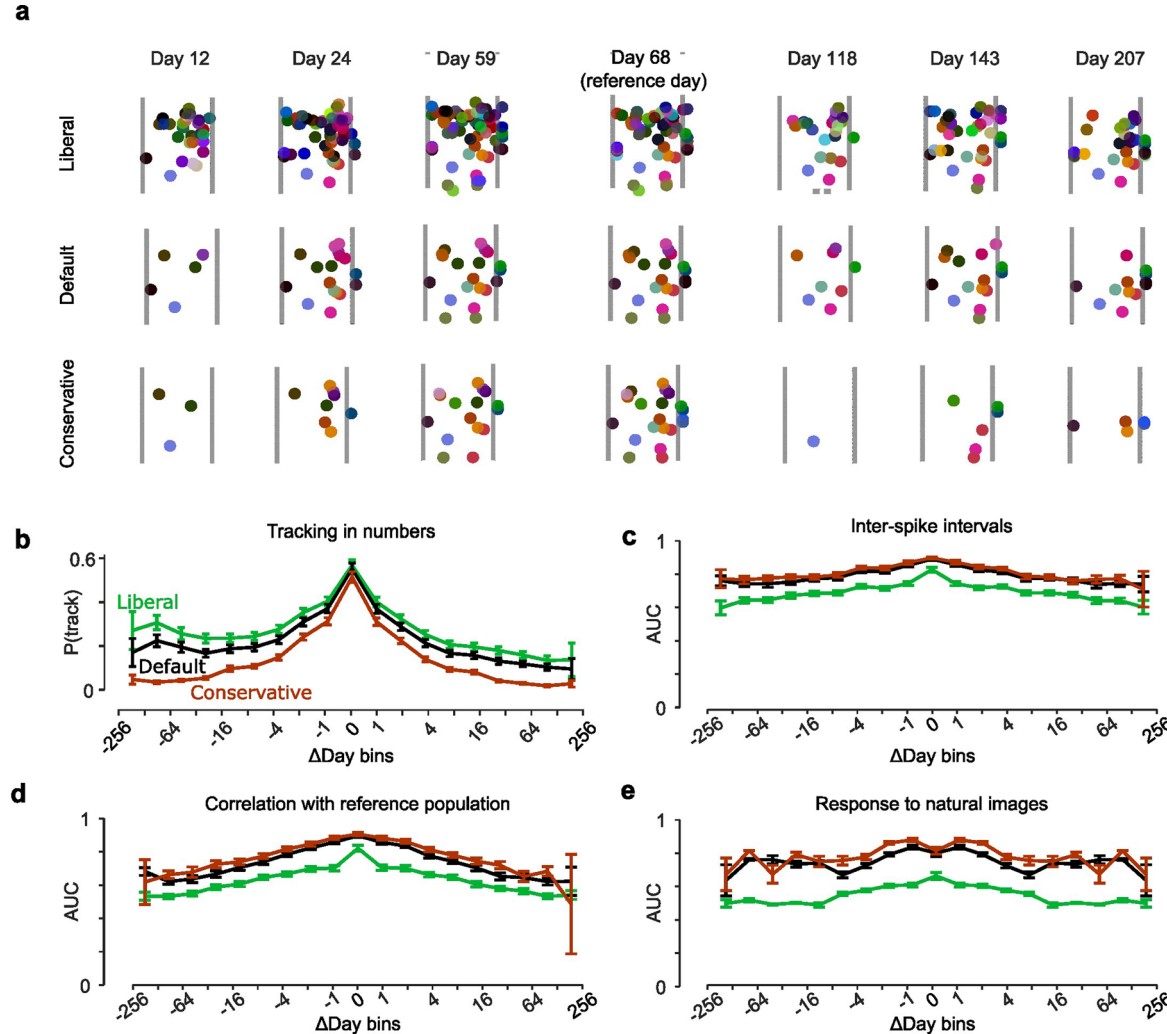

**Extended Data Fig. 7 | Tracking neurons over multiple sessions. a**, Example population of neurons that was tracked over 195 days in example mouse (ID1) – after drift correction. Other neurons on the probe are not shown. We compare the liberal, default, and conservative algorithm. Note that on some days neurons can disappear, to reappear on another day. **b**, Average ± s.e.m. probability of tracking a unit as a function of days in between recordings: the number of tracked neurons was divided by the total number of neurons available in the future recording (negative ΔDays) or past recording (positive ΔDays). We compare the liberal (green), default (black) and conservative (red) algorithms. Number of datasets per bin indicated in Fig. 5g (also applies to c-e). **c**, Average ± s.e.m area under the curve (AUC) values for inter-spike-interval histogram correlations when comparing matches versus non-matches as a function of number of days in between for liberal (green), default (black) and conservative (red) algorithms. **d**, same as **c**, but for the correlation with a reference population. **e**, same as **c**, but for responses to natural images.

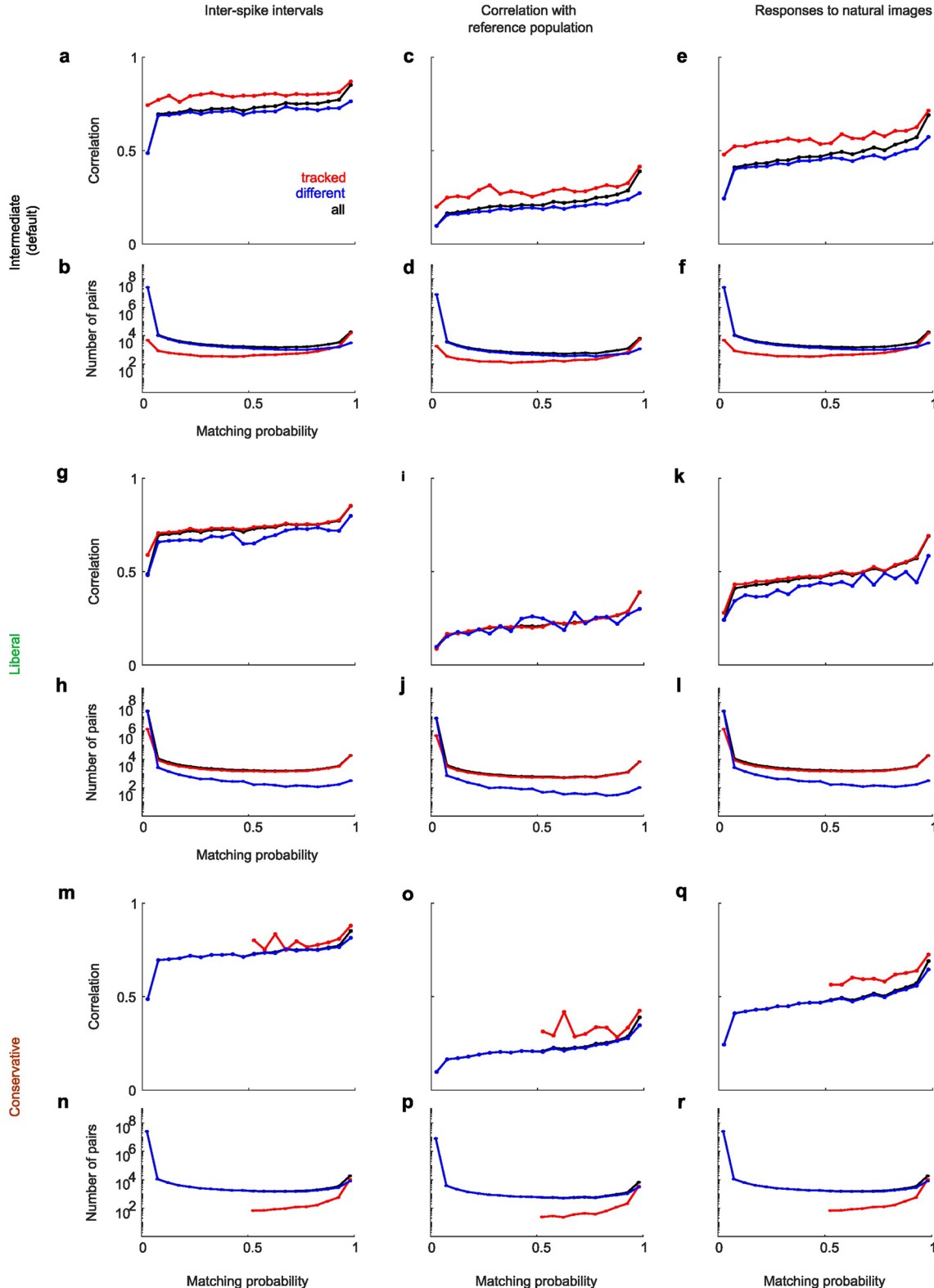

**Extended Data Fig. 8 | Relationship between match probability and functional property similarity. a**, Average correlation of the inter-spike interval across all pairs (*black*), pairs of clusters tracked as the same unit with the default algorithm (*red*), or different units (*blue*), as a function of the match probability for that pair (with bin size 0.05). One example animal is shown, and other animals showed similar patterns. **b**, Number of pairs per bin. **c-d**, Same as **a-b**, but for the correlation with a reference population. **e-f**, Same as **a-b**, but for the responses to natural images. **g-l**, Same as **a-f**, but using the liberal version of the algorithm. **m-r**, Same as **a-f**, but using the conservative version of the algorithm.

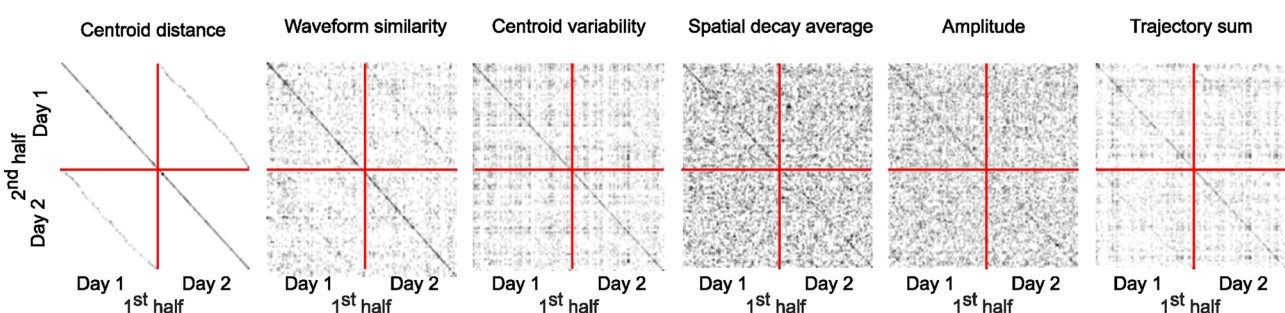

**Extended Data Fig. 9 | Similarity scores. a**, Area under the curve (AUC) for a receiver operating characteristic (ROC) classifying same units versus neighboring units. Large AUC values indicate that the similarity score is very informative in telling whether two waveforms come from the same unit versus whether that is not the case. Data points are 189 unique chronic recording datasets across 25 mice. **b**, Cross-correlation between each pair of similarity scores for example mouse. The diagonal shows histograms for the individual scores. When two parameters were very informative (large AUC scores) but correlated, we averaged them together (for example, waveform similarity is the average of waveform MSE and waveform correlations). **c**, Individual similarity scores for example mouse, sorted by depth shank and then depth. Thresholded at same prior as total score. Slightly smoothed to increase visibility (by averaging over 3 nearest neighbors over both columns and rows).

**Extended Data Table 1 | Information on animals, implants, and software used**

| ID | IMPLANT | PROBE | AREAS | EX | 4E,O | 4J | 5 | 6 | E2 | E4 | E5 | E6A | E7B-E | E9A |
|---|---|---|---|---|---|---|---|---|---|---|---|---|---|---|
| 1 | Cemented | Npx 2.0 (4s) | V1, HC | x | x | x | x | | x | x | x | x | x | x |
| 2 | Ultralight | Npx 1.0 (3B) | DMS | | x | | x | x | x | x | | x | x | x |
| 3 | Apollo | Npx 2.0 (4s) x2 | RSP, SC | | x | | x | | x | x | | x | x | x |
| 4 | Apollo | Npx 2.0 (4s) | V1, HC, TH | x | | | x | | x | x | | x | x | x |
| 5 | Apollo | Npx 2.0 (4s) | V1, HC | | x | x | x | | x | x | x | x | x | x |
| 6 | Cemented | Npx 1.0 (3B) | V1, HC | x | | | x | | | x | | | x | x |
| 7 | Cemented | Npx 2.0 (1s) | V1, HC | x | | | x | | | x | | | x | x |
| 8 | Cemented | Npx 2.0 (4s) | V1, HC | x | | x | x | | | x | x | | x | x |
| 9 | Repix | Npx 1.0 (3B) | V1, HC | | | | x | | | x | | | x | x |
| 10 | Repix | Npx 1.0 (3B) | V1, HC | x | | x | x | | | x | | | x | x |
| 11 | Repix | Npx 1.0 (3B) | V1, HC | x | | x | x | | | x | | | x | x |
| 12 | Repix | Npx 1.0 (3B) | V1, HC | | | | x | | | x | | | x | x |
| 13 | Haesler | Npx 1.0 (3B) | V1, HC | x | | x | x | | | x | | | x | x |
| 14 | Apollo | Npx 2.0 (4s) | V1, HC | x | | x | x | | | x | | | x | x |
| 15 | Apollo | Npx 2.0 (4s) | V1, HC | x | | x | x | | | x | x | | x | x |
| 16 | Apollo | Npx 2.0 (4s) | V1, HC | | | | x | | | x | | | x | x |
| 17 | Apollo | Npx 2.0 (4s) | V1, HC | x | | x | x | | | x | | | x | x |
| 18 | Apollo | Npx 2.0 (4s) | V1, HC | | | | x | | | x | | | x | x |
| 19 | Apollo | Npx 2.0 (4s) | V1, HC | | | | x | | | x | | | x | x |
| 20 | Apollo | Npx 2.0 (4s) x2 | FR, STR | x | | | x | | | x | x | | x | x |
| 21 | Apollo | Npx 2.0 (4s) x2 | FR, STR | x | | | x | | | x | | | x | x |
| 22 | Apollo | Npx 2.0 (4s) | FR | | | | x | | | x | | | x | x |
| 23 | Apollo | Npx 2.0 (4s) x2 | RSP, HC | | | | x | | | x | | | x | x |
| 24 | Apollo | Npx 2.0 (4s) | SC | | | | x | | | x | | | x | x |
| 25 | Apollo | Npx 2.0 (4s) x2 | DNS | | | | x | | | x | | | x | x |

For every animal (ID) we specify what type of implant was used, what probe type of Neuropixels (Npx) probe was used ('1s' and '4s' correspond to 1 and 4 shank Neuropixels probes respectively) and how many, and which areas the implant reached. The example mouse was ID1 (Figs. 2, 3, 4a–d; f–i; k–n; Extended Data Figs. 1, 3, 6b, 7a, 8, 9b, c). Abbreviations: V1: primary visual cortex, HC: hippocampus, DMS: dorsomedial striatum, RSP: retrosplenial cortex, SC: superior colliculus, TH: thalamus, FC: frontal cortex, STR: striatum. Mouse ID 1 was used for all example figures (Ex.) including Fig. 4a–d, f–i, k–n.

# Reporting Summary

## Statistics

For all statistical analyses, confirm that the following items are present in the figure legend, table legend, main text, or Methods section.

| n/a | Confirmed | |
|---|---|---|
| ☐ | ☒ | The exact sample size (*n*) for each experimental group/condition, given as a discrete number and unit of measurement |
| ☐ | ☒ | A statement on whether measurements were taken from distinct samples or whether the same sample was measured repeatedly |
| ☐ | ☒ | The statistical test(s) used AND whether they are one- or two-sided *Only common tests should be described solely by name; describe more complex techniques in the Methods section.* |
| ☒ | ☐ | A description of all covariates tested |
| ☐ | ☒ | A description of any assumptions or corrections, such as tests of normality and adjustment for multiple comparisons |
| ☐ | ☒ | A full description of the statistical parameters including central tendency (e.g. means) or other basic estimates (e.g. regression coefficient) AND variation (e.g. standard deviation) or associated estimates of uncertainty (e.g. confidence intervals) |
| ☐ | ☒ | For null hypothesis testing, the test statistic (e.g. *F*, *t*, *r*) with confidence intervals, effect sizes, degrees of freedom and *P* value noted *Give P values as exact values whenever suitable.* |
| ☐ | ☒ | For Bayesian analysis, information on the choice of priors and Markov chain Monte Carlo settings |
| ☒ | ☐ | For hierarchical and complex designs, identification of the appropriate level for tests and full reporting of outcomes |
| ☐ | ☒ | Estimates of effect sizes (e.g. Cohen's *d*, Pearson's *r*), indicating how they were calculated |

*Our web collection on statistics for biologists contains articles on many of the points above.*

## Software and code

Policy information about availability of computer code

**Data collection**    We have used SpikeGLX (https://billkarsh.github.io/SpikeGLX/) versions released on 20190413; 20190911; 20190912; 20200520; 20201012; 20201024; 20201103; 20230411 for Neuropixels data acquisition.

**Data analysis**    Analysis were done in Matlab R2023b and R2024. Neuropixels data were sorted using PyKilosort (https://github.com/MouseLand/pykilosort) and Kilosort 4 (https://www.nature.com/articles/s41592-024-02232-7) as indicated in the manuscript. Bombcell was used to define quality metrics of extracted units (https://doi.org/10.5281/zenodo.8172822) and define well isolated units. The rest of the analysis is described by this manuscript and can be found on Zenodo (https://zenodo.org/records/12734237) and Github (https://github.com/EnnyvanBeest/UnitMatch/tree/v1.0.0_UnitMatch).

For manuscripts utilizing custom algorithms or software that are central to the research but not yet described in published literature, software must be made available to editors and reviewers. We strongly encourage code deposition in a community repository (e.g. GitHub). See the Nature Portfolio guidelines for submitting code & software for further information.

## Data

Policy information about [availability of data](availability of data)

All manuscripts must include a [data availability statement](data availability statement). This statement should provide the following information, where applicable:

- Accession codes, unique identifiers, or web links for publicly available datasets
- A description of any restrictions on data availability
- For clinical datasets or third party data, please ensure that the statement adheres to our [policy](policy)

> Data for mouse ID 1-5 (Extended Data Table 1) are available via figshare (DOI: 10.6084/m9.figshare.24305758.v1) as part of the software demo. Further data is available via figshare (https://doi.org/10.5522/04/24411841.v1 ), and the rest can be made available upon request.

## Human research participants

Policy information about [studies involving human research participants and Sex and Gender in Research.](studies involving human research participants and Sex and Gender in Research.)

| | |
|---|---|
| Reporting on sex and gender | N/A |
| Population characteristics | N/A |
| Recruitment | N/A |
| Ethics oversight | N/A |

Note that full information on the approval of the study protocol must also be provided in the manuscript.

# Field-specific reporting

Please select the one below that is the best fit for your research. If you are not sure, read the appropriate sections before making your selection.

☒ Life sciences ☐ Behavioural & social sciences ☐ Ecological, evolutionary & environmental sciences

For a reference copy of the document with all sections, see [nature.com/documents/nr-reporting-summary-flat.pdf](nature.com/documents/nr-reporting-summary-flat.pdf)

# Life sciences study design

All studies must disclose on these points even when the disclosure is negative.

| | |
|---|---|
| Sample size | We have included data from 25 mice in this study. Specifics can be found in Table S1 of the manuscript. We used the data of five mice to build the method which is described in this manuscript. These data we have already published (https://doi.org/10.6084/m9.figshare.24305758.v1). Subsequently, we validated the method with the remaining data that we had recorded in the lab, and we have received feedback from other users that the described method works for their data as well. |
| Data exclusions | We included all available chronic datasets we had in the lab at the moment of performing the analysis. Exclusion criteria were based on individual recording quality: a minimum recording time (10 minutes);  a minimum number of well isolated units (25 units); a minimum number of  matches between recordings (20 matches). |
| Replication | Everyone can download the software and run the demo (https://github.com/EnnyvanBeest/UnitMatch/tree/v1.0.0_UnitMatch) with example data provided (https://doi.org/10.6084/m9.figshare.24305758.v1) or on their own data. |
| Randomization | n.a. mice were chronically implanted or not based on the needs for the specific study |
| Blinding | n.a. we were aware that mice were either chronically implanted or not. |

# Reporting for specific materials, systems and methods

We require information from authors about some types of materials, experimental systems and methods used in many studies. Here, indicate whether each material, system or method listed is relevant to your study. If you are not sure if a list item applies to your research, read the appropriate section before selecting a response.

## Materials & experimental systems

| n/a | Involved in the study |
|-----|----------------------|
| ☒ ☐ | Antibodies |
| ☒ ☐ | Eukaryotic cell lines |
| ☒ ☐ | Palaeontology and archaeology |
| ☐ ☒ | Animals and other organisms |
| ☒ ☐ | Clinical data |
| ☒ ☐ | Dual use research of concern |

## Methods

| n/a | Involved in the study |
|-----|----------------------|
| ☒ ☐ | ChIP-seq |
| ☒ ☐ | Flow cytometry |
| ☒ ☐ | MRI-based neuroimaging |

## Animals and other research organisms

Policy information about <u>studies involving animals</u>; <u>ARRIVE guidelines</u> recommended for reporting animal research, and <u>Sex and Gender in Research</u>

| | |
|---|---|
| Laboratory animals | Mice of Bl6 background. 3-9 months of age at implantation surgery. Implanted for max. 6 months. |
| Wild animals | No wild animals were used in the study. |
| Reporting on sex | Both males and females were used in this study. For this study sex is irrelevant. |
| Field-collected samples | No field collected samples were used in the study. |
| Ethics oversight | Experimental procedures were conducted at UCL according to the UK Animals Scientific Procedures Act (1986) and under personal and project licenses released by the Home Office following appropriate ethics review. |

Note that full information on the approval of the study protocol must also be provided in the manuscript.

