## [Peer Review File · Nature Methods]

Tracking neurons across days with high-density probes

Corresponding Author: Dr Enny van Beest

Version 0:

Decision Letter:

6th Feb 2024

Dear Dr van Beest,

Let me first sincerely apologize for the delay in the review process. Your Article, "Tracking neurons across days with high-density probes", has now been seen by two reviewers. As you will see from their comments below, although the reviewers find your work of considerable potential interest, they have raised a number of concerns. We are interested in the possibility of publishing your paper in Nature Methods, but would like to consider your response to these concerns before we reach a final decision on publication.

We therefore invite you to revise your manuscript to address these concerns. Specifically, reviewer #2 raises a number of concerns regarding the claims made and the advance over established methods. Please do make sure to thoroughly respond to these concerns. Please do not hesitate to contact me if you wish to discuss the revision.

Link Redacted

We hope to receive your revised paper within 2-3 months. If you cannot send it within this time, please let us know. In this event, we will still be happy to reconsider your paper at a later date so long as nothing similar has been accepted for publication at Nature Methods or published elsewhere.

OPEN SCIENCE REQUIREMENTS

REPORTING SUMMARY AND EDITORIAL POLICY CHECKLISTS

Reporting summary: <https://www.nature.com/documents/nr-reporting-summary.zip>
Editorial policy checklist: <https://www.nature.com/documents/nr-editorial-policy-checklist.zip>

DATA AVAILABILITY

All novel DNA and RNA sequencing data, protein sequences, genetic polymorphisms, linked genotype and phenotype data, gene expression data, macromolecular structures, and proteomics data must be deposited in a publicly accessible database, and accession codes and associated hyperlinks must be provided in the "Data Availability" section.

CODE AVAILABILITY

Please include a "Code Availability" subsection in the Online Methods which details how your custom code is made available. Only in rare cases (where code is not central to the main conclusions of the paper) is the statement "available upon request" allowed (and reasons should be specified).

For more information on our code sharing policy and requirements, please see: <https://www.nature.com/nature-research/editorial-policies/reporting-standards#availability-of-computer-code>

MATERIALS AVAILABILITY

ORCID

Best regards,
Nina

Nina Vogt, PhD
Senior Editor
Nature Methods

Reviewers' Comments:

Reviewer #1:

Remarks to the Author:

This paper addresses the tracking of single neurons in chronic electrophysiological recordings with high-density probes over long periods of time. An effective solution to this problem is very desirable as it opens up the possibility to monitor neural dynamics very precisely over extended periods, for instance to measure learning-related changes in neural activity that takes place over the course of days. Due to various experimental sources of instabilities, significant noise and variability in extracellular recordings, and the large size of the recorded data, this is a hard computational problem that can likely only be addressed with good statistical approaches. The paper demonstrates quite well that high-density arrays such as the Neuropixels probes (used here) provide sufficiently rich signals to compare recordings taken at different days with the same implanted probe.

The approach taken in this work is based on a set of useful features that are extracted for the putative neurons (units after spike sorting), and then compared in a pair-wise fashion between recordings. The first recording in a sequence is split in two halves to obtain the statistics of feature similarity scores for the matched/unmatched units. Similarity scores obtained for units from a pair of separate recordings are then classified according to these statistics. The performance of this approach is demonstrated using a whole range of complementary analysis, which includes manual curation (unfortunately currently still the gold standard) and physiological/functional properties of the recorded neurons. Overall I feel the authors have done what they could to demonstrate the approach is useful and works, in the absence of suitable ground truth data - I don't think it is really possible to generate realistic synthetic ground truth data for this situation, hence asking for such comparisons would be moot.

Despite the good demonstrated performance, this method will not be error-free - and hence understanding possible failure modes or having recommendations on how to interpret and validate the results is quite critical. This matters when especially tracking learning-related changes as shown in Figure 5, since erroneous matches can completely change the interpretation of such phenomena. Figure 4 shows that matched units have good functional stability over successive days, while Figure 5 only shows three example neurons - it would be really nice to see the population statistics also in the latter case, is it the case that many units drastically change (as implied by the population rates) while matching scores are similar? And generally, are there examples where there is a relation between functional unit similarity and feature matching score?

A further question I have is related to the method used to analyse several subsequent recordings. As far as I understand, in this case the first recording is taken as reference, and all subsequent recordings are compared against this. Given that instabilities accumulate over time, this would lead to a lower number of matched units discovered in recordings pairs that are more apart in time (and as shown in Figure S3). How does this compare to pairwise comparison of subsequent recordings? Are the matched units say between day 1 and 22 also part of the set of units matched between days 21 and 22? If this is the case, it would lend additional support to the method, and could perhaps even increase the unit yield over long periods of time.

I have looked at the code in the github repository and could run it using example data.

Reviewer #2:

Remarks to the Author:

In this manuscript, van Beest et al. approach the problem of tracking individual neurons across multi-day extracellular recordings. The authors introduce UnitMatch, an algorithm that tracks neurons based on properties of the waveform across multi-site pickup. The code for UnitMatch is available in a GitHub repository, and 2 d of recording data from 5 animals is available online. Briefly, the algorithm is: 1) spike sorting, 2) calculate the waveform in the first half and the second half of each session, 3) extract waveform parameters, 4) evaluate the similarity of all pairwise combinations of units (based on waveform metrics) across sessions 5) drift correction, 6) train a naive Bayes classifier based on 6 similarity metrics to estimate the posterior probability that clusters arise from the same neuron.

The authors then provide evidence that units matched across are functionally self-similar, as assessed by autocorrelogram, response to natural images, and correlation with a reference population. In long recordings (100 d) the authors continue to assess ROC curves in 2 d pairs - while this shows that neurons can be matched between, e.g., day 99 and 100, it does not (as the first impression suggests) reveal that data are presented from neurons tracked continuously across all 100 d. It seems that the big take away from the 100 d plots is that signal-to-noise ratio is stable over long recording in this laboratory - this would then intuitively enable UnitMatch to work as well on a pair of recordings in any point.

Similar algorithms and approaches to this problem present clear data showing the number of units tracked for various intervals of time. This number inevitably declines sharply with time. The authors acknowledge this briefly in the discussion (lines 319 to 324), in which they reveal that they are not presenting data from units tracked across months. This makes the presentation of stability across "183 days" feel a bit misleading.

Additional datasets are evaluated with UnitMatch - for example, the authors demonstrate that learning a wheel-rotation task does not disrupt the ability of unit tracking in extracellular recordings. The authors compare their results to concatenated datasets using Kilosort and a manually curated data. There is a surprising proportion of disagreement between the concatenated approach and UnitMatch, although the authors' explanation that algorithms have their own biases is reasonable. It is perhaps surprising that manual classification (match or no match) is highly effective as well.

UnitMatch is mathematically sound and the code works well. The algorithm is a valuable and rigorous but ultimately an incremental addition to relatively rich field with a plurality of open source options. The ability to track neurons over multiple recording segments is well documented, including Fraser and Schwartz, 2012; Hengen et al., 2016; Dhawale et al., 2017; Chung et al., 2018; Fraser and Schwartz, 2023 (similar to the present work, this focuses on neuropixels); Yuan et al., 2024 (biorxiv - this approach tracks neurons across multiple days, also utilizes high density silicon).

Author Rebuttal letter:

We thank the reviewers for their helpful suggestions. To address these suggestions and improve the work, we made a number of changes. The main ones are:

- True to the title of the paper, we now provide an algorithm to "track" neurons across multiple sessions, not just to "match" neurons across pairs of sessions, and we provide careful quantification of its performance (new section "Tracking over many recordings", and new Figure 5 and Figures S8 and S9).
- We now compare UnitMatch to the best existing algorithms for tracking, and we show that it greatly outperforms them in both speed and accuracy, (new section "Comparison with other methods", and new Figure S7).
- We now show that, during learning of a task, there is no relation between match probabilities and changes in functional properties (new panels in Figure 6)
- To further facilitate adoption, we have released a Python version of the pipeline (in addition to the previous MATLAB version), a Graphical User Interface to inspect the results, and an interface with the widely used spike-sorting environment SpikeInterface.

Reviewer 1

1. This paper addresses the tracking of single neurons in chronic electrophysiological recordings with high-density probes over long periods of time. An effective solution to this problem is very desirable as it opens up the possibility to monitor neural dynamics very precisely over extended periods, for instance to measure learning-related changes in neural activity that takes place over the course of days. Due to various experimental sources of instabilities, significant noise and variability in extracellular recordings, and the large size of the recorded data, this is a hard computational problem can likely only be addressed with good statistical approaches. The paper demonstrates quite well that high-density arrays such as the Neuropixels probes (used here) provide sufficiently rich signals to compare recordings taken at different days with the same implanted probe.

We thank the reviewer for this positive assessment.

2. The approach taken in this work is based on a set of useful features that are extracted for the putative neurons (units after spike sorting), and then compared in a pair-wise fashion between recordings. The first recording in a sequence is split in two halves to obtain the statistics of feature similarity scores for the matched/unmatched units. Similarity scores obtained for units from a pair of separate recordings are then classified

according to these statistics. The performance of this approach is demonstrated using a whole range of complementary analysis, which includes manual curation (unfortunately currently still the gold standard) and physiological/functional properties of the recorded neurons. Overall I feel the authors have done what they could to demonstrate the approach is useful and works, in the absence of suitable ground truth data - I don't think it is really possible to generate realistic synthetic ground truth data for this situation, hence asking for such comparisons would be moot.

We thank the reviewer for recognizing the efforts we made to demonstrate the robustness of our approach in absence of ground truth data. (A detail: as we now make clearer, UnitMatch splits in two halves all recordings in a dataset, not only the first recording).

3. Despite the good demonstrated performance, this method will not be error-free - and hence understanding possible failure modes or having recommendations on how to interpret and validate the results is quite critical.

We agree, and we now explain more clearly that UnitMatch is designed to be conservative in assigning matches, and we comment more explicitly on its biases (section "Performance metrics" and Figure S3a and S5). We also provide an algorithm for tracking neurons across multiple recordings (section "Tracking over many recordings", Figure 5, and Figure S8). Finally, our GitHub repository provides a script for users to do their own validation based on functional stability, and a GUI to inspect the matching pairs.

This matters when especially tracking learning-related changes as shown in Figure 5, since erroneous matches can completely change the interpretation of such phenomena. Figure 4 shows that matched units have good functional stability over successive days, while Figure 5 only shows three example neurons - it would be really nice to see the population statistics also in the latter case. Is it the case that many units drastically change (as implied by the population rates) while matching scores are similar?

We thank the reviewer for this suggestion. As we now explain more clearly, these recordings were conducted as a proof of principle, and yielded only a small population of neurons. With this limitation in mind, we have added a number of analyses in Figure 6: we now show the match probabilities and the inter-spike-interval histograms for the example neurons, and we plot match probability against changes in functional properties for all tracked neurons. These analyses show that, in this striatum dataset measured during learning, there is no relation between match probability and changes in functional properties for tracked neurons.

And generally, are there examples where there is a relation between functional unit similarity and feature matching score?

We thank the reviewer for this interesting question. As shown in Figure 4, the matches tended to have the highest functional similarity scores of all pairs, much higher than non-matching pairs. Furthermore, as we now explain (last paragraph of section "Tracking over many recordings" and new Figure S9) this was true even when the match probability of a tracked pair was 0 between a distant pair of recordings (but the algorithm had successfully tracked the unit across intermediate recordings). Overall, the functional similarity score for tracked units was similarly high across all match probabilities, and was much higher than

2

for units that were different from each other. This suggests that our algorithm successfully tracks neurons across many recordings.

4. A further question I have is related to the method used to analyse several subsequent recording. As far as I understand, in this case the first recording is taken as reference, and all subsequent recordings are compared against this. Given that instabilities accumulate over time, this would lead to a lower number of matched units discovered in recordings pairs that are more apart in time (and as shown in Figure S3). How does this compare to pairwise comparison of subsequent recordings? Are the matched units say between day 1 and 22 also part of the set of units matched between days 21 and 22? If this is the case, it would lend additional support to the method, and could perhaps even increase the unit yield over long periods of time.

We thank the reviewer for raising this point. We have now clarified the text on this point (first paragraph of Results): UnitMatch does not give a special role to the first recording in a sequence. Rather, UnitMatch uses the two halves of each recording, and compares the first half of all recordings against the second half of all recordings. It then outputs a match

probability for all pairs of units across all pairs of recordings, not just relative to the first recording (new Figure 5a and new Figures S6).

To help the users use this information, we now provide an algorithm for tracking units across multiple sessions (new Section "Tracking over many recordings" and new Figure 5 and Figures S8 and S9).

As for the yield of tracked neurons, on average it is indeed higher across sequential recordings than across more distant recordings (new Figure 5 and new Figures S6 and S8). However, as shown in Figure 5c, some neurons can disappear on one day (e.g. day 118) and reappear on a later day (e.g. day 143). This may be due to intermediate recordings in which the total neuron yield is low (e.g. because of noise).

I have looked at the code in the github repository and could run it using example data.

We thank the reviewer for testing our code. To further facilitate adoption, we have now released a Python version of the pipeline (in addition to the original MATLAB version), and an interface with the widely used spike-sorting environment SpikeInterface.

Reviewer 2

1. In this manuscript, van Beest et al. approach the problem of tracking individual neurons across multi-day extracellular recordings. The authors introduce UnitMatch, an algorithm that tracks neurons based on properties of the waveform across multi-site pickup. The code for UnitMatch is available in a GitHub repository, and 2 d of recording data from 5 animals is available online. Briefly, the algorithm is: 1) spike sorting, 2) calculate the waveform in the first half and the second half of each session, 3) extract waveform parameters, 4) evaluate the similarity of all pairwise combinations of units (based on waveform metrics) across sessions 5) drift correction, 6) train a naive Bayes classifier based

3

on 6 similarity metrics to estimate the posterior probability that clusters arise from the same neuron.

We thank the reviewer for the careful assessment of our manuscript.

2. The authors then provide evidence that units matched across are functionally self-similar, as assessed by autocorrelogram, response to natural images, and correlation with a reference population. In long recordings (100 d) the authors continue to assess ROC curves in 2 d pairs - while this shows that neurons can be matched between, e.g., day 99 and 100,

This is an important point, and we apologize for our lack of clarity. We now make it clearer that up to Figure 4 we only match units across pairs of recordings (which could be up to 235 days apart), and not across longer sequences of recordings. For instance, we clarify that the scores in the bottom two rows of Figure 4 were computed over pairs of days. We also added a new Figure S6 to show the number of matched neurons, and the corresponding functional validation, for each pair of recordings.

However, we agree that it is highly desirable to track neurons across multiple recordings (not just across two recordings). To this end, we now provide an algorithm for tracking units across multiple sessions (new Section "Tracking over many recordings" and new Figure 5 and Figure S6). This is a major improvement, and we thank the reviewer for encouraging it.

It seems that the big take away from the 100 d plots is that signal-to-noise ratio is stable over long recording in this laboratory - this would then intuitively enable UnitMatch to work as well on a pair of recordings in any point.

We agree. We now point this out 2nd paragraph of "Tracking over many recordings") and we point out another take away from Figure 4: that in all brain regions we recorded the functional fingerprints were remarkably stable over long periods of time – something that was not previously known. This means that functional fingerprints can be used to validate matches found by UnitMatch for recordings that are more than 100 days apart. More generally, we agree with the reviewer that the ability to track neurons with any software depends on the quality of the data, and we now discuss it explicitly (4th paragraph of discussion).

3. Similar algorithms and approaches to this problem present clear data showing the number of units tracked for various intervals of time. This number inevitably declines sharply with time. The authors acknowledge this briefly in the discussion (lines 319 to 324),

in which they reveal that they are not presenting data from units tracked across months. This makes the presentation of stability across “183 days” feel a bit misleading.

Indeed, as explained above the previous version of the manuscript only dealt with pairs of recordings, and showed that it could match neurons in pairs of recordings separated by 183 days (Figure 4). The new version clarifies this approach and adds the ability to track

4

neurons across multiple recordings (new Figure 5, and Figures S8 and S9). And indeed, we observe a decline of tracking probability over time, as expected. We thank the reviewer for encouraging this improvement.

Additional datasets are evaluated with UnitMatch – for example, the authors demonstrate that learning a wheel-rotation task does not disrupt the ability of unit tracking in extracellular recordings. The authors compare their results to concatenated datasets using Kilosort and a manually curated data. There is a surprising proportion of disagreement between the concatenated approach and UnitMatch, although the authors’ explanation that algorithms have their own biases is reasonable. It is perhaps surprising that manual classification (match or no match) is highly effective as well.

We understand the reviewer’s surprise that manual classification is effective. We now explain that manual curation was performed by six experts with great expense of time, and that the majority had to agree on a pair being a match for it to be considered a ‘curated match’ (4th paragraph of “Performance metrics”).

4. UnitMatch is mathematically sound and the code works well.

We thank the reviewer for testing our code. To further facilitate adoption, we have now released a Python version of the pipeline (in addition to the original MATLAB version), and an interface with the widely used spike-sorting environment SpikeInterface.

The algorithm is a valuable and rigorous but ultimately an incremental addition to relatively rich field with a plurality of open source options. The ability to track neurons over multiple recording segments is well documented, including Fraser and Schwartz, 2012; Hengen et al., 2016; Dhawale et al., 2017; Chung et al., 2018; Fraser and Schwartz, 2023 (similar to the present work, this focuses on neuropixels); Yuan et al., 2024 (biorxiv - this approach tracks neurons across multiple days, also utilizes high density silicon).

We now do a better job at explaining how the existing alternatives are inadequate for the purposes of tracking neurons across days with large and dense probes such as Neuropixels, and we touch on all the papers listed by the reviewer (except for Fraser and Schwartz 2023, which we could not locate).

As we explain in the Introduction (second to last paragraph), many existing methods rely on functional properties (we cite 12 papers doing this, including Fraser and Schwartz, 2012). However, if one uses functional properties to match neurons, one cannot then ask the key question of whether these measures change or remain constant.

As we explain in Discussion (second to last paragraph), some existing methods heavily depend on the recordings being continuous (Hengen et al., 2016; Dhawale et al., 2017; Chung et al., 2019). These algorithms have not been tested (or perform poorly) on sequences of recordings, or have not been tested with high-density electrodes. In the same paragraph we list six other reasons why our approach is more effective, including the fact that it operates after spike sorting, and that it computes match probabilities.

5

We also thank the reviewer for encouraging us to compare our study to that of Yuan et al., eLife, 2023. Our laboratory contributed data to that study, and we are well acquainted with it. We have now directly compared their algorithm with ours, by running them on the same data. The results (section “Comparison with other methods”, and new Figure S7) indicate that UnitMatch performs much better (and runs 8 time faster). To our knowledge, UnitMatch is therefore the best available tool for tracking neurons across recordings.

6

Decision Letter:

Our ref: NMETH-A54194A

28th May 2024

Dear Dr. van Beest,

Thank you for submitting your revised manuscript "Tracking neurons across days with high-density probes" (NMETH-A54194A). It has now been seen by the original referees and their comments are below. The reviewers find that the paper has improved in revision, and therefore we'll be happy in principle to publish it in Nature Methods, pending minor revisions to satisfy the referees' final requests and to comply with our editorial and formatting guidelines.

TRANSPARENT PEER REVIEW

ORCID

Best regards,
Nina

Nina Vogt, PhD
Senior Editor
Nature Methods

Reviewer #1 (Remarks to the Author):

An excellent revision, I have no additional comments.

Reviewer #2 (Remarks to the Author):

In response to the initial round of reviews, the authors have made several significant modifications that enhance the clarity, robustness, and utility of their manuscript. They have expanded their neuron tracking algorithm, UnitMatch, from matching neurons across pairs of sessions to tracking them over multiple sessions. These enhancements are well-documented in new sections of the manuscript, complete with additional figures. They also benchmarked UnitMatch against existing algorithms, establishing its superiority in both speed and accuracy in a new, dedicated section with supplemental figures.

In an effort to make their tools accessible, they will release a Python version of their pipeline, developed a graphical user interface, and integrated the tool with the widely used SpikeInterface. This is commendable.

They expand on the conservative nature of their matching approach, and the functional stability of matched neurons over time. While I am convinced that UnitMatch performs effectively and represents a technical refinement over existing methods, its

specialization for high-density silicon arrays may limit its general applicability. The community-wide ongoing effort to improve spike sorting technologies is simultaneously esoteric and beneficial for the field. UnitMatch contributes to this arms race. However, whether these improvements are sufficient for publication in this journal depends on the degree of novelty and practical impact they offer beyond incremental advances. That being said, this work is rigorous and well conducted.

Reviewer #2 (Remarks on code availability):

We reviewed the code in the first round. We have not revisited it in the second round.

Version 2:

Decision Letter:

3rd Sep 2024

Dear Dr van Beest,

I am pleased to inform you that your Article, "Tracking neurons across days with high-density probes", has now been accepted for publication in Nature Methods. The received and accepted dates will be October 18th, 2023 and September 3rd, 2024. This note is intended to let you know what to expect from us over the next month or so, and to let you know where to address any further questions.

Over the next few weeks, your paper will be copyedited to ensure that it conforms to Nature Methods style. Once your paper is typeset, you will receive an email with a link to choose the appropriate publishing options for your paper and our Author Services team will be in touch regarding any additional information that may be required. It is extremely important that you let us know now whether you will be difficult to contact over the next month. If this is the case, we ask that you send us the contact information (email, phone and fax) of someone who will be able to check the proofs and deal with any last-minute problems.

Please note that *Nature Methods* is a Transformative Journal (TJ). Authors may publish their research with us through the traditional subscription access route or make their paper immediately open access through payment of an article-processing charge (APC). Authors will not be required to make a final decision about access to their article until it has been accepted. [Find out more about Transformative Journals](https://www.springernature.com/gp/open-research/transformative-journals)

Best regards,
Nina

Nina Vogt, PhD
Senior Editor
Nature Methods

** Visit the Springer Nature Editorial and Publishing website at http://editorial-jobs.springernature.com?utm_source=ejP_NMeth_email&utm_medium=ejP_NMeth_email&utm_campaign=ejp_Nmeth for more information about our career opportunities. If you have any questions please click [here](mailto:editorial.publishing.jobs@springernature.com).

Open Access This Peer Review File is licensed under a Creative Commons Attribution 4.0 International License, which permits use, sharing, adaptation, distribution and reproduction in any medium or format, as long as you give appropriate credit to the original author(s) and the source, provide a link to the Creative Commons license, and indicate if changes were made. In cases where reviewers are anonymous, credit should be given to 'Anonymous Referee' and the source. The images or other third party material in this Peer Review File are included in the article's Creative Commons license, unless indicated otherwise in a credit line to the material. If material is not included in the article's Creative Commons license and your intended use is not permitted by statutory regulation or exceeds the permitted use, you will need to obtain permission directly from the copyright holder.
